



# Earth System Model Evaluation Tool (ESMValTool) v2.0 – diagnostics for emergent constraints and future projections from Earth system models in CMIP

Axel Lauer[1], Veronika Eyring[1,2], Omar Bellprat[3], Lisa Bock[1], Bettina K. Gier[2,1], Alasdair Hunter[5], Ruth Lorenz[4], Núria Pérez-Zanón[5], Mattia Righi[1], Manuel Schlund[1], Daniel Senftleben[1], Katja Weigel[2,1], and Sabrina Zechlau[1]

[1]Deutsches Zentrum für Luft- und Raumfahrt (DLR), Institut für Physik der Atmosphäre, Oberpfaffenhofen, Germany
[2]University of Bremen, Institute of Environmental Physics (IUP), Bremen, Germany
[3]Swiss Federal Department of Foreign Affairs, Bern, Switzerland
[4]ETH Zurich, Institute for Atmospheric and Climate Science, Zurich, Switzerland
[5]Barcelona Supercomputing Center (BSC), Barcelona, Spain

Correspondence to: Axel Lauer (axel.lauer@dlr.de)

## Abstract

The Earth System Model Evaluation Tool (ESMValTool), a community diagnostics and performance metrics tool for evaluation and analysis of Earth system models (ESMs) is designed to facilitate a more comprehensive and rapid comparison of single or multiple models participating in the coupled model intercomparison project (CMIP). The ESM results can be compared against observations or reanalysis data as well as against other models including predecessor versions of the same model. The updated and extended version 2.0 of the ESMValTool includes several new analysis scripts such as large-scale diagnostics for evaluation of ESMs as well as diagnostics for extreme events, regional model and impact evaluation. In this paper, the newly implemented climate metrics such as effective climate sensitivity (ECS) and transient climate response (TCR) as well as emergent constraints for various climate-relevant feedbacks and diagnostics for future projections from ESMs are described and illustrated with examples using results from the well-established model ensemble CMIP5. The emergent constraints implemented include ECS, snow-albedo effect, climate-carbon cycle feedback, hydrologic cycle intensification, future Indian summer monsoon precipitation, and year of disappearance of summer Arctic sea ice. The diagnostics included in ESMValTool v2.0 to analyze future climate projections from ESMs include analysis scripts to reproduce selected figures of chapter 12 of the Intergovernmental Panel on Climate Change's (IPCC) Fifth Assessment report (AR5) and various multi-model statistics.



# 1    Introduction

Climate models are important tools not only to improve our understanding of the key processes in present-day climate but also to project future climate change under different plausible scenarios. Climate models have been continuously improved and extended over the last decades from relatively simple atmosphere-only models to the
complex state-of-the-art Earth system models (ESMs) participating in the latest (sixth) phase of the Coupled Model Intercomparison Project (CMIP6, Eyring et al. (2016a)). The increasing complexity of the models is needed to represent key feedbacks that affect climate change, but is also likely to increase the spread of climate projections across the multi-model ensemble (Eyring et al., 2019). This poses a challenge for evaluation and analysis of the model results that requires efficient tools able to handle the increasing number of variables,
processes and also the increasing data volume.

The ESMValTool released in a first version in 2016 (Eyring et al., 2016b) has been developed with the aim of taking model evaluation to the next level by facilitating analysis of many different ESM components, providing well-documented source code and scientific background of implemented diagnostics and metrics and allowing for traceability and reproducibility of results (provenance). This has been made possible by a lively and growing
development community continuously improving the tool supported by multiple national and European projects. The release of version 2.0 (v2.0) of the ESMValTool that is documented in this and accompanying papers (Eyring et al., in revision; Righi et al., accepted; Weigel et al., in prep.) has been developed as a large community effort to specifically target the increased data volume of CMIP6 and the related challenges posed by analysis and evaluation of output from multiple high-resolution and complex ESMs. For this, the core functionalities have
been completely rewritten in order to take advantage of state-of-the-art computational libraries and methods to allow for efficient and user-friendly data processing (Righi et al., accepted). The new version of the ESMValTool now also includes diagnostics for emergent constraints and for analysis of future projections from ESMs that are described in this article. Additionally, v2.0 includes the two new climate metrics effective climate sensitivity (ECS) and transient climate response (TCR) (Sect. 3.2).

An emergent constraint is a relationship across an ensemble of models between some aspect of the Earth system sensitivity and an observable trend or variation in the current climate, which offers the possibility to reduce uncertainties in climate projections. Furthermore, emergent constraints can help guiding model development onto processes crucial to the magnitude and spread of future climate change projections and to point out future observational priorities (Eyring et al., 2019). Emergent constraints implemented in ESMValTool v2.0 (Sect. 3.3)
include seven different approaches to constrain ECS as well as constraints for the hydrological cycle intensification, snow-albedo effect, year of disappearance of summer Arctic sea ice, future Indian summer monsoon precipitation and climate-carbon cycle feedback.



For the analysis of ESM projections, ESMValTool v2.0 now includes diagnostics to reproduce selected figures from chapter 12 (Long-term Climate Change: Projections, Commitments and Irreversibility) of the

Intergovernmental Panel on Climate Change's (IPCC) Fifth Assessment report (AR5) (Collins et al., 2013). These include figures showing the change in a variable between historical and future periods, e.g. maps (2D variables), zonal means (3D variables), time series showing the change in certain variables from historical to future periods for multiple scenarios, and maps visualizing change in variables normalized by global mean temperature change (pattern scaling) and the possibility to show statistical significance of changes when

compared to natural variability and the degree of agreement between the models using the stippling and hatching methods as in Collins et al. (2013). Furthermore, diagnostics tailored to analyze projections of sea ice such as, for example, calculation of the year of disappearance (sea ice extent below 1 million $km^2$) from a multi-model ensemble and to constrain the future austral jet position have been added. A newly implemented "toy model" can be used to generate synthetic members of a single dataset. When providing an estimate for the standard error of

observations e.g. from differences between different observational datasets, this toy model can be used to investigate and take into account the effect of observational uncertainty in model evaluation (Sect. 3.4). A summary is given in Sect. 4. The aim of this paper is to document and illustrate how these newly added ESMValTool "recipes", i.e. configuration files defining input, preprocessing, diagnostics and run-time options of the ESMValTool, can be used for model evaluation and analysis.

## 2   Models and observations

The open-source release of ESMValTool (v2.0) that accompanies this paper is intended to work with CMIP5 and CMIP6 model output (and partly also with CMIP3 if the required output has been saved), but the tool is compatible with any arbitrary model output, provided that it is in CF-compliant (Climate and Forecast) netCDF format and that the variables and metadata are following the CMOR (Climate Model Output Rewriter) tables and

definitions. Observations used in the evaluation are detailed in the various sections of the manuscript (see also Section 6) and summarized in Table 1 and Table 2 but should also be seen as examples as they can be easily replaced by other observational datasets provided they follow the CMOR convention. For selected observational datasets, reformat scripts are provided with the ESMValTool that contain detailed downloading and processing instructions to convert the datasets into a CMOR-like format that can be processed by the ESMValTool. These

reformat scripts serve as examples for writing similar scripts for other observational datasets that do not follow the CMOR standard.



## 3 Overview of recipes included in ESMValTool v2.0 for emergent constraints and future projections

In this section, all diagnostics and metrics newly added to the ESMValTool v2.0 for analysis of future projections from ESMs as well as the emergent constraints implemented are described and illustrated with
examples using results from the CMIP5 model ensemble (Taylor et al., 2012). The ESMValTool workflow is controlled by configuration files called "recipes", which define all input datasets, pre-processing steps and diagnostics to run (for details we refer to Righi et al. (accepted)). An overview of all recipes described in this paper including a short description, the variables processed, the names of the diagnostic scripts and observations is given in Table 1.

### 3.1 Calculations of multi-model products

Multi-model means are commonly used to project climate change (IPCC, 2013, 2007) and are thus a useful quantity to calculate in support of diagnostics included in the ESMValTool.

The recipe *recipe_multimodel_products.yml* computes the multi-model ensemble mean for a set of models selected by the user for individual variables and different temporal resolutions (annual, seasonal, monthly). After
selecting the region (rectangular region defined by the lowermost and uppermost longitudes and latitudes), the mean for the selected reference period is subtracted from the time series in order to obtain the anomalies for the desired period. In addition, the recipe computes the percentage of models agreeing on the sign of this anomaly, thus providing some information on the robustness of the climate change signal.

The output of the recipe consists of a contour map showing the time average of the multi-model mean anomalies
and stippling to indicate locations where the percentage of models agreeing on the sign of the multi-model mean anomaly exceeds a threshold selected by the user (Figure 1). The example in Figure 1 shows a warming over the continents in the range of 1-2 K which is more pronounced than the warming over the ocean which is mostly in the range of 0.5-1.5 K in this scenario. The example also shows that the models largely agree on the sign of the temperature change with the most prominent exceptions found in parts of the Southern Ocean, Greenland and the
North Atlantic. Furthermore, a time series of the area-weighted mean anomalies is plotted. For the plots, the user can select the length of the running window for temporal smoothing and choose to display the ensemble mean with a light shading to represent the spread of the ensemble or choose to display each individual model (Figure 2). The example in Figure 2 shows an increase in global average June temperatures up to about 2060 when temperatures start to level off. By 2100 the four CMIP5 example models MPI-ESM-MR, CNRM-CM5, BCC-
CSM1-1 and IPSL-CM5A-LR show a spread in temperature increase for the RCP2.6 scenario ranging from 0.7 K to about 1.8 K.





### 3.2 Effective climate sensitivity (ECS) and transient climate response (TCR)

The effective climate sensitivity (ECS) is an important metric to assess the future warming of the climate system. It is defined as the change in global mean near-surface air temperature as a result of a doubling of the atmospheric $CO_2$ concentration compared to pre-industrial conditions after the climate system has reached a new equilibrium (Gregory et al., 2004). Climate models of the CMIP5 model ensemble simulated an ECS ranging between 2.1 and 4.7 K (Flato et al., 2013). Using all available evidence of that time, IPCC AR5 assessed a "likely" range of ECS between 1.5 and 4.5 K in 2013 (IPCC, 2013). *recipe_ecs.yml* uses a regression method proposed by Gregory et al. (2004) to calculate ECS. Using the total radiative forcing F caused by the doubling of atmospheric $CO_2$ concentration and the climate feedback parameter $\lambda$, ECS is defined as $ECS = F / \lambda$. Both of these variables can be assessed by linear regression of the equation for radiative balance $N = F - \lambda \, \Delta T$, where N is the net radiation flux at the top of the atmosphere (TOA) and $\Delta T$ the global mean near-surface air temperature change. N and $\Delta T$ are both given as global and annual mean differences between the abrupt four times $CO_2$ simulation and the linear regression of the pre-industrial control run. Figure 3 illustrates this regression for the CMIP5 multi-model mean. Moreover, it shows that the assumption of a linear climate feedback parameter is only an approximation. Using only the first 20 years (last 130 years) instead of all 150 years of the abrupt four times $CO_2$ simulations results in a stronger (weaker) feedback, which again leads to a lower (higher) ECS. This demonstrates the different response of the climate system at different timescales, i.e. non-linear feedback processes. This diagnostic requires the input variables near-surface air temperature (tas), TOA incoming shortwave radiation (rsdt), TOA outgoing shortwave radiation (rsut) and TOA outgoing longwave radiation (rlut) from abrupt4xCO2 (quadrupling of $CO_2$ compared to pre-industrial conditions) and piControl (pre-industrial control) simulations.

Figure 9.42a of Flato et al. (2013) shows the globally averaged mean near-surface air temperature (GMSAT) for the historical period 1961-1990 plotted vs. ECS of several CMIP5 models. The latter quantity can be calculated by a regression method based on Gregory et al. (2004) as outlined above. A similar figure produced with *recipe_flato13ipcc.yml* implemented in ESMValTool v2.0 shows that there are no distinctive correlations between the historical surface temperatures and the ECS, which suggests that the ECS is not very sensitive to errors in the current climate in contrast to other sources of uncertainty (Figure 4).

The transient climate response (TCR) is defined as the global and annual mean near-surface air temperature anomaly in the 1pctCO2 simulation (1% increase in $CO_2$ per year) for a 20-year period centered at the time of $CO_2$ doubling, i.e. using the years 61 to 80 after the start of the simulation. The temperature anomalies are calculated by subtracting a linear fit to the piControl run for all 140 years from the 1pctCO2 experiment prior to the TCR calculation (Gregory and Forster, 2008). Figure 5 shows (a) a time series of the 1pctCO2 near-surface



temperature anomalies from MIROC-ESM used to obtain TCR and (b) TCR values for different CMIP5 models

calculated with *recipe_tcr.yml*.

### 3.3     Emergent constraints

An emergent constraint utilizes an ensemble of ESMs together with observational data to constrain a simulated future Earth system feedback. A prerequisite for an emergent constraint is a robust relationship between, for example, changes occurring on seasonal or interannual time scales and changes found in ESM simulations of

anthropogenically-forced climate change (Eyring et al., 2019). If such a relationship can be explained by a plausible physical mechanism, an observational constraint of multi-model projections of quantities that cannot be observed directly might be possible. Such a non-observable quantity is, for instance, ECS. The technique of emergent constraints offers the possibility to reduce uncertainties in climate projections and can help guiding model development by highlighting processes that are crucial to explaining the magnitude and spread of the

modeled future climate change. Emergent constraints can also help pointing out the need for more and/or more reliable observations. Table 2 summarizes the emergent constraints that have been implemented in ESMValTool (v2.0) including the observational datasets used and are described in the following.

### 3.3.1     Emergent constraints on effective climate sensitivity

*recipe_ecs_scatter.yml* calculates five emergent constraints for ECS (see Table 2). These are briefly described in

the Sections 3.3.1.1 to 3.3.1.5. The ECS values from the models are pre-calculated with *recipe_ecs.yml* (see Section 3.2) or can be taken from literature. The diagnostic calculates ECS vs. selected constraining parameters such as, for instance, the climatological Hadley cell extent from models, and fits a linear regression line to the data. If available, the observational uncertainty of a given observational dataset can be estimated. For this, the standard error of the observations is subtracted or added from or to the means before calculating the

observational value (estimated minimum or maximum, respectively). In addition to the scatter plots of ECS vs. constraining parameter calculated by the diagnostic, the diagnostic also outputs the 25% / 75% confidence intervals of the regression (i.e. uncertainty of the fit) and the 25% / 75% prediction intervals of the regression (i.e. measure for the quality of the linear fit). By definition, 50% of all model data points are within the 25% / 75% prediction interval of the regression line. Examples of the different scatterplots that can be created by

*recipe_ecs_scatter.yml* are shown in Figure 6. It should be noted that because a different set of CMIP5 models might be used in the figures compared to the originally published emergent constraints, the figures could show some deviations to the ones published in literature. While the emergent constraints shown in Figure 6a,c,d,e suggest ECS values in the upper range of the values given in IPCC AR5 ((IPCC) (2007), 1.5 to 4.5 K), the emergent constraint shown in Figure 6b suggests an ECS value in the lower range of the IPCC AR5 values.



In addition to these five emergent constraints, *recipe_cox18nature.yml* implements an emergent constraint for ECS based on global temperature variability (Section 3.3.1.6), *recipe_ecs_multivariate_constraint_cmip5.yml* an emergent constraint based on the difference between tropical and mid-latitude cloud fraction (Section 3.3.1.7).

### 3.3.1.1   Covariance of shortwave cloud reflection

This emergent constraint uses the models' correlation of the covariance of tropical low-level cloud (TLC)
reflection with the underlying SST to constrain ECS (Brient and Schneider, 2016). The definition and calculation of the individual terms follows Brient and Schneider (2016): TLC regions are defined as the 25% ocean areas between 30°S and 30°N with the lowest 500-hPa relative humidity. TLC reflection is calculated as the ratio of top of the atmosphere shortwave cloud radiative forcing and insolation, both averaged over the TLC region. This is then used to calculate the regression coefficients of deseasonalized variations of TLC shortwave reflection and
sea surface temperature in % per K used as emergent constraint. In the example shown in Figure 6a, data from the CMIP5 historical simulations between 1980 and 2005 are used for the models, observational / reanalysis data used in Figure 6 are ERA-Interim (Dee et al., 2011) for relative humidity, HadISST (Rayner et al., 2003) for sea surface temperatures, and CERES-EBAF (Ed2.7) (Loeb et al., 2012) for top of the atmosphere radiative fluxes.

### 3.3.1.2   Climatological Hadley cell extent

Lipat et al. (2017) found that the climatological mean Hadley cell (HC) edge latitude from CMIP5 models correlates with ECS. The HC edge latitude is calculated from first two grid cells from the equator going south where the zonal average 500-hPa mass stream function changes sign from negative to positive (downward branch of the HC). The mass stream function is calculated from climatological December-January-February (DJF) means of the meridional wind fields. The correlation of the climatological HC extent with ECS found in
CMIP5 models is explained by observations that show a correlation of variability in mid-latitude clouds and cloud radiative effects with poleward HC expansion (Lipat et al., 2017). For the example shown in Figure 6b, CMIP5 data from historical simulations and ERA-Interim (Dee et al., 2011) as reference dataset for the years 1980-2005 are used.

### 3.3.1.3   Lower tropospheric mixing index

Following Sherwood et al. (2014), the lower tropospheric mixing index (LTMI) can be used to constrain ECS and is calculated as the sum of small-scale mixing S and the large-scale component of mixing D. S is calculated from relative humidity (RH) and temperature (T) differences between 700 and 850 hPa and averaged over a tropical region between 30°S and 30°N defined by the upper quartile of the annual mean 500-hPa ascent rate within ascending regions: $S = (\Delta RH_{700-850}/100\% - \Delta T_{700-850}/9K) / 2$. The large-scale component of mixing is the
ratio of shallow to deep overturning: $D = \langle \Delta H(\Delta)H(-\omega_1)\rangle / \langle-\omega_2 H(-\omega_2)\rangle$ with $\omega_1$ the average of the vertical velocity at 850 and 700 hPa, $\omega_2$ the average of the vertical velocity at 600, 500, and 400 hPa, H the step function, and $\langle...\rangle$ the average over the tropical ocean region 160°W-30°E, 30°S-30°N. The lower tropospheric mixing



index is calculated as LTMI = S + D. Sherwood et al. (2014) explain the correlation between LTMI and ECS in CMIP3 and CMIP5 models by convective mixing between the lower and middle tropical troposphere

dehydrating low-level cloud layers at an increasing rate as climate warms. They argue that this rate of increase depends on initial mixing strength, which links the mixing to clouds feedbacks and thus ECS. Figure 6c shows an example of this emergent constraint applied to CMIP5 historical simulations using ERA-Interim data (Dee et al., 2011) as reference data. All datasets in this example cover the time period 1980-2005.

### 3.3.1.4    Southern ITCZ index

The southern ITCZ index (Bellucci et al., 2010; Hirota et al., 2011) is defined as the climatological annual mean precipitation bias averaged over the south-eastern Pacific (30°S-0°, 150°W-100°W) given in mm day$^{-1}$. The southern ITCZ index is used to quantify the double-ITCZ bias in CMIP3 and CMIP5 models and has been found to correlate with ECS (Tian, 2015). In the example shown in Figure 6d, the ITCZ index has been calculated from CMIP5 historical model simulations averaged over the years 1980-2005. TRMM (Huffman et al., 2007) satellite

data (v7) averaged over the years 1998-2013 have been used as observational reference.

### 3.3.1.5    Tropical mid-tropospheric humidity asymmetry index

The strong link found in CMIP3 and CMIP5 models between the double-ITCZ bias and simulated moisture, precipitation, clouds, and large-scale circulation allows the double-ITCZ bias and thus ECS to also be related to mid-tropospheric humidity over the tropical Pacific (Tian, 2015). As shown by Tian (2015), spatial patterns of

mid-tropospheric humidity and precipitation are similar as both are related to the ITCZ. This allows defining a tropical mid-tropospheric humidity asymmetry index to quantify the double-ITCZ bias in models and consequently constrain ECS. This index is defined as relative bias in simulated annual mean 500-hPa specific humidity compared with observations ((model – observation) / observation * 100%) averaged over the Southern Hemisphere (SH) tropical Pacific (30°S-0°, 120°E-80°W) minus the bias averaged over the Northern

Hemisphere (NH) tropical Pacific (20°N-0°, 120°E-80°W) (Tian, 2015). The example for the tropical mid-tropospheric humidity asymmetry index shown in Figure 6e is calculated from CMIP5 historical runs averaged over the years 1980-2005 and AIRS (v5) satellite data (Susskind et al., 2006) averaged over the years 2003-2010 as observational reference data.

### 3.3.1.6    Global temperature variability

Cox et al. (2018) propose an emergent constraint for the ECS using global temperature variability. The latter is defined by a metric $\psi$ which can be calculated from the global temperature variance (in time) $\sigma_T$ and the one-year-lag autocorrelation of the global temperature $\alpha_{1T}$ by

$$\psi = \frac{\sigma_T}{\sqrt{-\ln(\alpha_{1T})}}.$$

Using the simple "Hasselmann model" (Hasselmann, 1976), Cox et al. (2018) showed that $\psi$ is linearly correlated with ECS in CMIP5 data. Since calculation of $\psi$ only depends on the temporal evolution of the global




surface temperature, there are many observational datasets available. In the original publication, data from
HadCRUT4 (Morice et al., 2012) are used to construct the emergent relationship. In the ESMValTool, this is
reproduced by *recipe_cox18nature.yml*, which only needs the two variables historical near-surface air
temperature (tas) and ECS (see Section 3.2). The emergent relationship between ECS and ψ is shown in Figure 7
including means and confidence intervals. The constrained range of ECS based on this plot is 2.2 K to 3.4 K with
a 66% confidence interval, similar to Cox et al. (2018).

### 3.3.1.7    Difference between tropical and mid-latitude cloud fraction

Volodin (2008) proposes an emergent constraint for ECS based on the distribution of clouds in global climate
models. The study finds that models with high climate sensitivity show a higher total cloud cover over the
southern mid-latitudes and a lower total cloud cover over the tropics than the multi-model average. Thus, the
difference in tropical total cloud cover (between 28°S and 28°N) and the SH mid-latitude total cloud cover
(between 56°S and 36°S) is negatively correlated with ECS. The original publication uses the CMIP3 ensemble
and the ISCCP-D2 dataset (Rossow and Schiffer, 1991) as observational reference, but the relationship also
holds when using CMIP5 models. In the ESMValTool, this emergent constraint for ECS can be produced with
*recipe_ecs_multivariate_constraint_cmip5.yml*, which uses CMIP5 historical runs averaged between 1980 and
2000 (Figure 8). The observed values are based on ISCCP-D2 data and are taken from Volodin (2008).

### 3.3.2    Emergent constraints on the carbon cycle

Uncertainties in projections of future temperature using ESMs are high, in a large part due to uncertainties of
emissions and feedbacks. Within the carbon-cycle, feedbacks are usually split into the carbon cycle – climate
feedback γ, which quantifies carbon to climate change, and the carbon cycle – $CO_2$ concentration feedback β,
which is the carbon sensitivity to atmospheric $CO_2$ (Friedlingstein et al., 2006). γ is a positive feedback as
climate warming reducing the efficiency of $CO_2$ absorption by the land and ocean, leading to more of the emitted
carbon staying in the atmosphere which in turn leads to additional warming. In constrast, β is a negative
feedback because of the so-called $CO_2$ fertilization effect, where plants take up a higher amount of $CO_2$ for
photosynthesis with increasing atmospheric $CO_2$ concentrations. Efforts have been made to reduce the
uncertainties of these two carbon cycle feedback parameters.

Wenzel et al. (2014) employed the emergent constraint described by Cox et al. (2013) for the long-term
sensitivity of tropical land carbon storage to climate warming ($γ_{LT}$) to the interannual sensitivity of atmospheric
$CO_2$ to interannual tropical temperature variability ($γ_{IAV}$) in CMIP5 models. The analysis from this paper can be
reproduced using *recipe_wenzel14jgr.yml* with the emergent relationship being able to reduce the range of
projected $γ_{LT}$ (Figure 9). Input variables include net primary productivity (nbp), surface temperature (tas), gas
exchange flux of $CO_2$ into the ocean (fgco2) from the experiment 1pctCO2, nbp, fgco2, tas from the emission




driven historical simulations (esmHistorical), as well as nbp from the esmFixClim1 (carbon cycle sees $CO_2$ concentration increase, but radiation doesn't) simulations. The different simulations are included in $\gamma_{IAV}$, which is estimated from both, the 1pctCO2 experiment as well as the esmHistorical simulation, and then compared in

the paper. The default observational datasets are NCEP reanalysis (Kalnay et al., 1996) for the surface temperature and the global carbon project (GCP; Le Quere et al. (2015)) for the carbon fluxes.

Wenzel et al. (2016a) developed an emergent constraint for β on land in the extratropics and northern mid-latitudes constraining the projected land photosynthesis with changes in the seasonal cycle of atmospheric $CO_2$. The figures from this paper can be reproduced with *recipe_wenzel16nat.yml*, with Figure 10 showing the

emergent constraint reproduced with the ESMValTool. The unconstrained $CO_2$ fertilization effect lies at 40 ± 20%, which can be narrowed down to 37 ± 9% in high-latitudes and 32 ± 9% in the extratropics with this emergent constraint. Input variables from the models needed to run this recipe is gross primary productivity (gpp) in the esmFixClim1 simulations, as well as the atmospheric $CO_2$ concentration (co2) from emission driven historical simulations. Observations used are the atmospheric $CO_2$ concentrations at Point Barrow (BRW;

156.6°W, 71.3°N), Alaska and Cape Kumukahi, Hawaii (KMK; 155.6°W, 19.5°N) (NOAA, 2018).

### 3.3.3    Emergent constraints on the year of disappearance of September Arctic sea ice

This sea ice diagnostic produces scatterplots of (a) mean of and (b) trend in historical Arctic September sea ice extent (SSIE) vs. first year of disappearance (YOD). Here, YOD is defined as the first of five consecutive years in which the Arctic SSIE drops below one million km² (Wang and Overland, 2009). Sea ice extent is defined in

the diagnostic as the total area of all grid cells in which the sea ice concentration is 15% or larger, Arctic is defined as the region north of 60°N. The annual minimum Arctic sea ice extent typically occurs in September. For this reason, September mean sea ice quantities are commonly used in literature for analyses of the timing of an ice-free Arctic (e.g., Massonnet et al. (2012); Sigmond et al. (2018)). The two scatterplots (Figure 11a) and (Figure 11b) are similar to figures 12.31 a/c of Collins et al. (2013), respectively. In addition, the diagnostic

produces a scatterplot of mean SSIE vs. trend in historical SSIE, similar to figure 2 of Massonnet et al. (2012). In the example shown in Figure 11, HadISST data (Rayner et al., 2003) over the time period 1960-2005 have been used as reference dataset for comparison with CMIP5 results. The figure shows that while the individual models spread widely around the observed mean Arctic SSIE, most of the CMIP5 models tend to underestimate the trend in Arctic SSIE observed over the period 1960-2005.

### 3.3.4    Emergent constraints on the snow-albedo effect

The recipe *recipe_snowalbedo.yml* computes springtime snow-albedo feedback values in climate change vs. springtime values in the seasonal cycle in transient climate change experiments following Hall and Qu (2006).



The strength of the snow-albedo effect is quantified by the variation in net incoming shortwave radiation (Q) with surface air temperature ($T_s$) due to changes in surface albedo $\alpha_s$:

$$\left(\frac{\partial Q}{\partial T_s}\right) = -I_t \cdot \frac{\partial \alpha_p}{\partial \alpha_s} \cdot \frac{\Delta \alpha_s}{\Delta T_s}$$

Here, $I_t$ is the constant incoming solar radiation at the top of the atmosphere, $\alpha_p$ the planetary albedo. The diagnostic produces scatterplots of simulated springtime $\Delta\alpha_s$ / $\Delta T_s$ values in climate change (ordinate) vs. simulated springtime $\Delta\alpha_s$ / $\Delta T_s$ values in the seasonal cycle (abscissa). These values are calculated as follows: (ordinate values) the change in April $\alpha_s$ (future projection - historical) averaged over NH land masses poleward of 30°N is divided by the change in April $T_s$ (future projection - historical) averaged over the same region. The

change in $\alpha_s$ (or $T_s$) is defined as the difference between 22nd century mean $\alpha_s$ ($T_s$) and 20th-century-mean $\alpha_s$. Values of $\alpha_s$ are weighted by April incoming insolation ($I_t$) prior to averaging.

(Abscissa values) the seasonal cycle $\Delta\alpha_s$ / $\Delta T_s$ values, based on 20th century climatological means, are calculated by dividing the difference between April and May $\alpha_s$ (averaged over NH continents poleward of 30°N) by the difference between April and May $T_s$ averaged over the same area. Values of $\alpha_s$ are weighted by

April incoming insolation prior to averaging.

Figure 12 shows an example calculated from CMIP5 historical (1901-2000) and Representative Concentration Pathways 4.5 (RCP4.5, 2101-2200) experiments for 12 different models. The seasonal cycle values used as reference (vertical gray line) are calculated from third generation of ISCCP radiative fluxes (ISCCP-FH, Young et al. (2018)) and near-surface air temperature from ERA-Interim (Dee et al., 2011) for the years 1984-2000.

While data from ISCCP-FH data suggest that CMIP5 models tend to underestimate springtime snow-albedo effect values in climate change, using the second generation of ISCCP radiative fluxes (ISCCP-FD, Zhang et al. (2004), not shown) as in figure 9.45a of Flato et al. (2013) suggest that the CMIP5 models under- and overestimate springtime snow-albedo effect almost equally.

### 3.3.5    Emergent constraints on the hydrological cycle

The recipes *recipe_deangelis2015nat.yml* and *recipe_li2017natcc.yml* newly developed for v2.0 reproduce the analysis from DeAngelis et al. (2015) and Li et al. (2017), respectively. DeAngelis et al. (2015) constrain the hydrologic cycle intensification with observed radiative fluxes and water vapor data. The recipe *recipe_deangelis2015nat.yml* reproduces their figures 1b (Figure 13a) to 4 (Figure 13b) as well as their extended data figures 1 and 2. So far the analysis is available for 21 CMIP5 models and includes monthly mean total

precipitable water on a 1 x 1 degree grid from RSS (Remote Sensing System) Version-7 microwave radiometer data (Wentz et al., 2007) and ERA-Interim reanalysis (Dee et al., 2011), as well as radiative fluxes from the dataset Clouds and the Earth's Radiant Energy System Energy Balance and Filled (CERES-EBAF, Kato et al.



(2013); Loeb et al. (2009)). Figure 13a shows that energy sources and sinks readjust in reply to an increase in greenhouse gases, leading to a decrease in the sensible heat flux and an increase in the other fluxes; Figure 13b

shows that results from parameterization schemes using pseudo-k-distributions with more than 20 exponential terms representing water vapor absorption and correlated-k-distributions agree better with the observations than the other schemes.

Li et al. (2017) relate the future Indian summer monsoon projections to the present-day precipitation over the tropical western Pacific. With this relationship they can correct the projected rainfall for models with too strong

negative cloud–radiation feedback on sea surface temperature. The corrected values (see Figure 14) do not show an increase in rainfall over the whole ISM region under greenhouse warming and are expected to be more robust than the uncorrected projection (Li et al., 2017). The *recipe_li2017natcc.yml* reproduces their figures 1 and 2 for an ensemble of 22 CMIP5 models (Figure 14) and their figure 1a for each of the individual models and the multi-model mean.

**3.4    Climate model projections**

In addition to the emergent constraints described in the previous section, ESMValTool v2.0 also includes new diagnostics specifically designed to analyze future climate projections from ESMs. This includes diagnostics using the multiple diagnostic ensemble regression used to constrain the future position of the austral jet, a "toy model" to allow for investigating the effect of observational uncertainty on model evaluation, diagnostics for

reproducing selected figures from the climate projection chapter in IPCC AR5 (Collins et al., 2013) and for analyzing future sea ice quantities. All of these new diagnostics in ESMValTool v2.0 are briefly described in the following sections.

**3.4.1    MDER to constrain future austral jet position**

The position of the austral jet stream is poorly modeled by CMIP5 models with a latitude range of 10° within the

ensemble and a mean bias towards the equator. The *recipe_wenzel16jclim.yml* reproduces the study of Wenzel et al. (2016b) who used a process-oriented multiple diagnostic ensemble regression (MDER) to constrain the future jet position in the RCP4.5 scenario. MDER uses a stepwise regression scheme to identify the most relevant present-day diagnostics from a list of diagnostics provided as an input and links those to future projections via a multivariate linear regression scheme. With the diagnostics selected by MDER, the future quantity (in this case

the austral jet position) can be constrained with suitable observationally based data (here: ERA-Interim (Dee et al., 2011)), following the same basic idea as emergent constraints (see also section 3.3). Using this approach, the future jet position from CMIP5 models is bias-corrected about 1.5° southwards compared to the unweighted multi-model mean (Figure 15).





### 3.4.2 Toy model

Synthetic datasets generated from "toy models" have been used in the literature for assessing the effectiveness of multi-model combination strategies and for estimating the effect of observational uncertainties on the correlation between forecasts and observational datasets (Massonnet et al., 2016). The Toy model recipe implemented into ESMValTool v2.0 is based on the approach presented in Weigel et al. (2008) for simulating single-model ensembles from a Gaussian distribution, where the number of members and the standard deviation of the error

are defined by the user. Following Weigel et al. (2008), the recipe takes as input a set of observations, $y_1$, $y_2$, …, $y_N$, and for each observation $y_i$, M synthetic members x are generated from:

$$x_{i,m} = \alpha y_i + \epsilon_\beta + \epsilon_{i,m}$$

where $y \sim N(0,1)$, $\epsilon_\beta \sim N(0,\beta)$ and $\epsilon_1,...,\epsilon_M \sim N(0,\sqrt{(1-\alpha^2-\beta^2)})$ with the notation $N(\mu,\sigma)$ referring to a random number drawn from a normal distribution with mean $\mu$ and standard deviation $\sigma$. The simulated value $x_m$ is

obtained by multiplying y by $\alpha$, the predictability of the observation, which is set to 1 in this instance, and by adding perturbations $\epsilon_{i,m}$ and $\epsilon_\beta$. The simulated values have the following properties:

1) The simulated values $x_{i,1}$ have the same climatology as the observations.

2) The mean correlation between the simulations $x_{i,1}$,…., $x_{i,M}$ and observation $y_i$ is determined by $\alpha$.

3) The parameter $\beta$ describes the model under-dispersion, where $\beta = 0$ corresponds to the case where the

synthetic ensemble is well dispersed and covers the full range of uncertainties for a given correlation $\alpha$. The under-dispersion increases with $\beta$.

Here, the predictability $\alpha$ is 1 since we are only interested in generating synthetic observations. Thus, the user only needs to define the standard deviation of the error. This term can be based on the observational uncertainty when available (e.g. as provided with the European Space Agency's Climate Change Initiative (ESA CCI) SST

dataset; Merchant et al. (2014a); Merchant et al. (2014b)) or estimated by the user, e.g. by estimating the standard deviation between different observational reference datasets (Bellprat et al., 2017). For further discussion of this synthetic value generator, its general application to forecasts and its limitations, see Weigel et al. (2008). The recipe *recipe_toymodel.yml* writes a netCDF file containing the synthetic observations. Due to the sampling of the perturbations from a Gaussian distribution, running the recipe multiple times, with the same

observation dataset and input parameters, will result in different outputs (Figure 16).

### 3.4.3 Climate projection chapter of IPCC WGI AR5

The *recipe_collins13ipcc.yml* reproduces a subset of the figures from the long-term climate change projections chapter of the IPCC AR5 (Chapter 12, Collins et al. (2013)). This new recipe in version 2.0 allows for reproduction of selected figures from AR5 to show changes between historical and future projections over the

available CMIP models. It will also allow a faster analysis of the CMIP6 climate projections that are part of the





Scenario Model Intercomparison Project (ScenarioMIP, O'Neill et al. (2016)). The recipe includes figures such as time series from historical periods to projections (including spread among models, see Figure 17), horizontal maps for individual models as well as multi-model means (including stippling and hatching to indicate significant changes and areas where models do not agree, see Figure 18), and vertical zonal mean plots (also

including stippling and hatching to indicate significant changes). The example shown in Figure 18 shows where the CMIP5 models project an increase in precipitation and where they project a decrease. This example also shows quite large regions where the projections are still uncertain (hatching).

Most diagnostics scripts are set up in a generic way, so that in principle they can be used for any variable from the CMIP archive. The scripts have been tested for the variables indicated in Table 1. To be able to determine if

a change signal is larger than natural variability the natural variability is calculated from the piControl runs, other than that the recipe uses historical and RCP runs. All diagnostics in this recipe with the exception of the emergent constraints on the year of disappearance of September Arctic sea ice (Section 3.3.3) do not use observations.

### 3.4.4    Sea ice

The sea ice diagnostics included in the ESMValTool (*recipe_seaice.yml*) have been extended with three new diagnostics. The first new diagnostic *seaice_trends.ncl* calculates the trend in sea ice extent or sea ice area from each model and reference observation(s) or reanalysis data that are given in the recipe. The diagnostic produces histogram plots of the trend distributions from all models and adds the reference datasets (here: HadISST, Rayner et al. (2003)) as colored vertical lines. The user can specify the region (Arctic or Antarctic) and the

month of the year for which sea ice area/extent is calculated. The trends are calculated over the full period specified in the recipe and the resulting plots are similar to Flato et al. (2013) Figures 9.24 c/d. The example plot (Figure 19) shows that the majority of CMIP5 models slightly underestimate the observed trend in summer sea ice extent over the time period 1960-2005.

The second new diagnostic *seaice_yod.ncl* calculates the year of near-disappearance of Arctic sea ice. The

diagnostic creates a time series plot of September Arctic sea ice extent for each model given in the recipe and adds three multi-model statistics: mean, standard deviation and YOD. It optionally reads a list of pre-calculated model weights and adds the weighted multi-model mean time series including weighted multi-model standard deviation to the plot (see for example figure 7 of Senftleben et al. (2020)). The example in Figure 20 shows that there is a large spread in simulated sea ice extent among the CMIP5 models with individual models simulating a

summer sea ice extent below 1 million km$^2$ already around the year 2025 while other models are still well above this threshold in 2100.





The third new diagnostic *seaice_ecs.ncl* calculates emergent constraints for YOD using mean or trend in sea ice extent. The diagnostic produces scatter plots of different historical and future sea ice metrics, similar to figure 2 of Massonnet et al. (2012) and figures 12.31 a/c of Collins et al. (2013) (see Section 3.3.3 for details).

**4    Summary**

This paper is part of a series of four articles describing the new features and diagnostics of the Earth System Model Evaluation Tool v2.0. Version 2.0 is a major upgrade from the last release v1.1.0 (Eyring et al., 2016b; Lauer et al., 2017). Besides many technical improvements including greatly improved performance and user-friendliness (Righi et al., accepted), version 2.0 includes new large-scale diagnostics for evaluation of Earth

system models (Eyring et al., in revision) and diagnostics for extreme events, regional model and impact evaluation and analysis of ESM results (Weigel et al., in prep.). In this article, newly implemented diagnostics and metrics to analyze projections from ESMs and emergent constraints for climate-relevant parameters including effective climate sensitivity, snow-albedo effect, climate-carbon cycle feedback, hydrologic cycle intensification, future Indian summer monsoon precipitation, land photosynthesis and year of disappearance of

summer Arctic sea ice are described and illustrated with examples using CMIP5 data.

The implemented multi-model products (*recipe_multimodel_products.yml*) allow for an easy and quick overview of the multi-model ensemble mean and the inter-model agreement in the sign of the multi-model mean anomaly for a given variable, geographical region, season and time period. In addition to maps showing the anomalies and their inter-model agreement, the results are also given as anomaly time series showing each individual model and

the multi-model ensemble mean, which can be used to estimate the inter-model spread.

Effective climate sensitivity and transient climate response can be calculated for a model ensemble with *recipe_ecs.yml* and *recipe_tcr.yml*, respectively. Both, ECS and TCR, are climate metrics that can be used to estimate and compare the sensitivity the simulated near-surface temperature from individual models to increased atmospheric $CO_2$ concentrations. With these metrics, it is possible to group the models in high- and low-

sensitivity models for further analysis.

Emergent constraints offer the possibility to use an ensemble of ESMs together with observations in order to constrain non-observable parameters such as simulated future Earth system feedbacks. Seven emergent constraints are available in ESMValTool v2.0 for ECS (*recipe_ecs_scatter.yml*, *recipe_cox18nature.yml* and *recipe_ecs_multivariate_constraint_cmip5.yml*): (1) covariance of shortwave cloud reflection using the models'

correlation of the covariance of tropical low-level cloud reflection with the underlying SST (Brient and Schneider, 2016); (2) latitude of the climatological mean Hadley cell edge (Lipat et al., 2017); (3) atmospheric convective mixing calculated as sum of small- and large-scale component, the lower tropospheric mixing index (Sherwood et al., 2014); (4) bias in climatological annual mean precipitation over the south-eastern Pacific, the



southern ITCZ index (Tian, 2015); mid-tropospheric humidity over the tropical Pacific, the tropical mid-
tropospheric humidity asymmetry index (Tian, 2015); (6) global temperature variability (Cox et al., 2018); and
(7) difference between tropical and mid-latitude cloud fraction (Volodin, 2008). Two emergent constraints on the
hydrological cycle are implemented: (1) a constraint on the hydrological cycle intensification that uses
observations of radiative fluxes and water vapor (DeAngelis et al., 2015) in *recipe_deangelis2015nat.yml*; and
(2) a constraint on the future Indian summer monsoon using present-day precipitation data over the tropical
western Pacific (Li et al., 2017) implemented in *recipe_li2017natcc.yml*. Additionally, emergent constraints are
available for the carbon cycle: (1) future tropical land carbon storage (Wenzel et al., 2014),
*recipe_wenzel14jgr.yml*; (2) projected land photosynthesis (Wenzel et al., 2016a), *recipe_wenzel16nat.yml*. Also
implemented are emergent constraints for the year of disappearance of September Arctic sea ice (Massonnet et
al., 2012) in *recipe_seaice.yml* and for the snow-albedo effect (Hall and Qu, 2006) (*recipe_snowalbedo.yml*).

Various new diagnostics are available specifically for analysis of climate model projections. The multiple
diagnostic ensemble regression (MDER) method has been implemented to constrain the projected position of the
austral jet following Wenzel et al. (2016b). The method uses a stepwise regression to identify the most relevant
diagnostics (calculated with present-day data) that are linked to projections of a quantity via a multivariate linear
regression scheme. Observational data can then be used to constrain the projected quantity such as the future
austral jet position (*recipe_wenzel16jclim.yml*).

A number of newly implemented diagnostics resembling selected figures from IPCC AR5 chapter 12 (Collins et
al., 2013) for analysis of climate model projections are grouped in *recipe_collins13ipcc.yml*. The diagnostics
include time series and horizontal and vertical zonal maps including stippling and hatching to show significant
changes between a climate projection scenario and a historical simulation. For the stippling and hatching, results
from pre-industrial control runs are used to estimate internal variability of a variable, which is then used to assess
whether simulated changes are significant or not. In *recipe_seaice.yml*, diagnostics to analyze sea ice in climate
model simulations are grouped. The new diagnostics include calculation of trends in sea ice area and extent,
multi-model estimates for the year of disappearance of sea ice in climate projections, and scatter plots of
different historical and future sea ice metrics such as historical trend in sea ice extent vs. YOD. In addition, a
"toy model" (*recipe_toymodel.yml*) has been implemented into ESMValTool v2.0 that allows generating
synthetic ensemble members from a single dataset (Weigel et al., 2008). When applied to observational data, this
can be used to take into account observational uncertainty when comparing the observations with model results.
For this, the user needs to specify the standard error of the observations that is provided with some observational
datasets or estimated from differences between different observational datasets for the same quantity.

ESMValTool v2.0 has been specifically developed in order to analyze and evaluate the latest generation of
CMIP model results. The new version is now available to the community for scientific analyses of CMIP6 data.



The ESMValTool development team will continue to improve and extend the tool. The ongoing ESMValTool development and discussions regarding new features can be followed on GitHub at https://github.com/ESMValGroup. Feedback, bug reports and contributions by the scientific community are very

welcome at any time.

## 5    Code availability

ESMValTool v2.0 is released under the Apache License, VERSION 2.0. The latest release of ESMValTool v2.0 is publicly available on Zenodo at https://doi.org/10.5281/zenodo.3401363. The source code of the ESMValCore package, which is installed as a dependency of the ESMValTool v2.0, is also publicly available on Zenodo at

https://doi.org/10.5281/zenodo.3387139. ESMValTool and ESMValCore are developed on the GitHub repositories available at https://github.com/ESMValGroup.

## 6    Data availability

CMIP5 data are available freely and publicly from the Earth System Grid Federation (ESGF). Observations used in the evaluation are detailed in the various sections of the manuscript. The observational datasets are not

distributed with the ESMValTool that is restricted to the code as open source software. Observational datasets that are available through the Observations for Model Intercomparisons Project (obs4MIPs, https://esgf-node.llnl.gov/projects/obs4mips/) can be downloaded freely from the ESGF and used directly with the ESMValTool. For all other observational datasets, the ESMValTool provides a collection of scripts (NCL and Python) with exact downloading and processing instructions to recreate the datasets used in this publication.

**Author contribution**

AL and VE coordinated the ESMValTool v2.0 diagnostic effort and led the writing of the paper. MR helped coordinating the diagnostic implementation and testing in ESMValTool v2.0. All other authors contributed individual diagnostics to this release. All authors contributed to the text.

**Competing interests**

The authors declare that they have no conflict of interest.

**Acknowledgements**

The development of ESMValTool (v2.0) is supported by several projects. The diagnostic development of ESMValTool v2.0 for this paper was supported by different projects with different scientific focus, in particular by (1) European Union's Horizon 2020 Framework Programme for Research and Innovation "Coordinated



Research in Earth Systems and Climate: Experiments, kNowledge, Dissemination and Outreach
(CRESCENDO)" project under Grant Agreement No. 641816, (2) Copernicus Climate Change Service (C3S)
"Metrics and Access to Global Indices for Climate Projections (C3S-MAGIC)" project, (3) Federal Ministry of
Education and Research (BMBF) CMIP6-DICAD project, (4) ESA Climate Change Initiative Climate Model
User Group (ESA CCI CMUG), and (5) Helmholtz Society project "Advanced Earth System Model Evaluation
for CMIP (EVal4CMIP)". In addition, we received technical support on the ESMValTool v2.0 development
from the European Union's Horizon 2020 Framework Programme for Research and Innovation "Infrastructure
for the European Network for Earth System Modelling (IS-ENES3)" project under Grant Agreement No 824084.
We acknowledge the World Climate Research Program's (WCRP's) Working Group on Coupled Modelling
(WGCM), which is responsible for CMIP, and we thank the climate modelling groups for producing and making
available their model output. We thank Franziska Winterstein (DLR) for her helpful comments on the
manuscript. The computational resources of the Deutsches Klimarechenzentrum (DKRZ, Germany) were
essential for developing and testing this new version and are kindly acknowledged.

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





Table 1 Overview of recipes for emergent constraints and future projections implemented in ESMValTool (v2.0) along with the section they are described, a brief description, the required CMIP5 variables, the diagnostic scripts included and the observational datasets used in the examples. For further technical details, we refer to the GitHub repository.

| Recipe name | Section | Description | Variables | Diagnostic scripts | Observational datasets |
|---|---|---|---|---|---|
| **Section 3.1 Calculations of multi-model products** | | | | | |
| recipe_multimodel_products.yml | 3.1 | tool to compute the ensemble mean anomaly, ensemble variance and agreement and plot the results as maps and time series | tas (example) | magic_bsc/multimodel_products.r | - |
| **Section 3.2 Effective climate sensitivity (ECS) and transient climate response (TCR)** | | | | | |
| recipe_ecs.yml | 3.2 | ECS using linear regression following Gregory et al. (2004) | rtmt, rtnt, tas | climate_metrics/ecs.py | - |
| recipe_flato13ipcc.yml | 3.2 | Figure 9.42 of Flato et al. (2013): (a) global mean near-surface air temperature vs. ECS; (b) TCR vs. ECS | rtmt, rtnt, tas | climate_metrics/ecs.py climate_metrics/tcr.py ipcc_ar5/ch09_fig09_42a.py ipcc_ar5/ch09_fig09_42b.py | - |
| recipe_tcr.yml | 3.2 | transient climate response (TCR) following Gregory and Forster (2008) | tas | climate_metrics/tcr.py | - |
| **Section 3.3 Emergent constraints** | | | | | |
| recipe_ecs_scatter.yml | 3.3.1.1-3.3.1.5 | ECS vs. different quantities (Brient and Schneider, 2016; Lipat et al., 2017; Sherwood et al., 2014; Tian, 2015) | hur, hus, pr, rsdt, rsut, rsutcs, ta, ts, va, wap | emergent_constraints/ecs_scatter.ncl | ERA-Interim (hur, ta, va, wap), TRMM (pr), AIRS (hus), HadISST (ts), CERES-EBAF (rsdt, rsut, rsutcs) |
| recipe_cox18nature.yml | 3.3.1.6 | emergent constraint for ECS based on global temperature variability following Cox et al. (2018) | tas, tasa | climate_metrics/ecs.py climate_metrics/psi.py emergent_constraints/cox18nature.py | HadCRUT4 (tas, tasa) |
| recipe_ecs_multivariate_constraint_cmip5.yml | 3.3.1.7 | ECS vs. difference between tropical and mid-latitude cloud fraction (Volodin, 2008) | clt | emergent_constraints/ecs_scatter.py | ISCCP-D2 (clt) |
| recipe_wenzel14jgr.yml | 3.3.2 | emergent constraint on long-term sensitivity of tropical land carbon storage to climate warming ($\gamma_{LT}$) (Wenzel et al., 2014) | fgco2, nbp, tas | carbon_ec/carbon_constraint.ncl carbon_ec/carbon_gammaHist.ncl carbon_ec/carbon_tsline.ncl | NCEP (tas), GCP (nbp, fgco2) |





| recipe_wenzel16nat. yml | 3.3.2 | emergent constraint on carbon cycle - $CO_2$ concentration feedback ($\beta$) (Wenzel et al., 2016a) | gpp, co2 | carbon_ec/carbon_beta. ncl carbon_ec/carbon_cycl e_co2.ncl carbon_ec/carbon_co2-gpp-correlation.ncl | NOAA station measurements Alaska and Hawaii (co2) |
|---|---|---|---|---|---|
| recipe_seaice.yml | 3.3.3 | emergent constraint on YOD following Massonnet et al. (2012) | sic, areacello | seaice/seaice_ecs.ncl | HadISST (sic) |
| recipe_snowalbedo. yml | 3.3.4 | emergent constraint on snow-albedo effect following Hall and Qu (2006) | rsdscs, rsdt, rsuscs, tas | emergent_constraints/s nowalbedo.ncl | ISCCP-FH (alb, rsdt), ERA-Interim (tas) |
| recipe_deangelis15n at.yml | 3.3.5 | constraint on hydrologic cycle intensification (DeAngelis et al., 2015) | hfss, lvp, prw, rlnst, rlnstcs, rsnst, rsnstcs, rsnstcsnor m, tas | deangelis15nat/deangel is1b.py deangelis15nat/deangel is2.py deangelis15nat/deangel is3.py | ERA-Interim (prw), RSS (prw), CERES-EBAF (rlnstcs, rsnst, rsnstcs, rsnstcsnorm) |
| recipe_li2017natcc.y ml | 3.3.5 | emergent constraint on the future Indian summer monsoon precipitation following Li et al. (2017) | pr ,ts, ua, va | emergent_constraints/li f1.py | GPCP (pr) |
| **Section 3.4 Climate model projections** | | | | | |
| recipe_wenzel16jcli m.yml | 3.4.1 | constraint on austral jet position in future projections | asr, ps, ta, uajet (ua), va | austral_jet/asr.ncl austral_jet/main.ncl mder/absolute_correlati on.ncl mder/regression_stepwi se.ncl mder/select_for_mder.n cl | ERA-Interim (ps, ta, ua, va), CERES-EBAF (asr) |
| recipe_toymodel.ym l | 3.4.2 | recipe for generating synthetic observations based on the model presented in Weigel et al. (2008) | psl (example) | magic_bsc/toymodel.R | ERA-Interim (psl) |





| recipe_collins13ipcc.yml | 3.4.3 | selected figures from IPCC AR5, chap. 12 (Collins et al., 2013): mainly difference maps between future and present | areacello, clt, evspsbl, hurs, mrro, mrsos, pr, psl, rlut, rsut, rtmt, sic, snw, sos, ta, tas, thetao, ua | ipcc_ar5/ch12_calc_IAV_for_stippandhatch.ncl ipcc_ar5/ch12_calc_map_diff_mmm_stippandhatch.ncl ipcc_ar5/ch12_calc_zonal_cont_diff_mmm_stippandhatch.ncl ipcc_ar5/ch12_map_diff_each_model_fig12-9.ncl ipcc_ar5/ch12_plot_map_diff_mmm_stipp.ncl ipcc_ar5/ch12_plot_ts_line_mean_spread.ncl ipcc_ar5/ch12_plot_zonal_diff_mmm_stipp.ncl ipcc_ar5/ch12_snw_area_change_fig12-32.ncl ipcc_ar5/ch12_ts_line_mean_spread.ncl emergent_constraints/snowalbedo.ncl | HadISST (sic) |
|---|---|---|---|---|---|
| recipe_seaice.yml | 3.4.4 | time series of sea ice area and extent, ice extent trend distributions, year of near disappearance of Arctic sea ice, emergent constraint on YOD (Massonnet et al., 2012) | areacello, sic | seaice/seaice_aux.ncl seaice/seaice_ecs.ncl seaice/seaice_trends.ncl seaice/seaice_tsline.ncl seaice/seaice_yod.ncl | HadISST (sic) |





Table 2 Emergent constraints implemented in ESMValTool v2.0 and observational datasets used.

| Reference | Constrained Parameter | Description / observed quantity | Observational datasets |
|---|---|---|---|
| Brient and Schneider (2016) | ECS | covariance of shortwave cloud reflection | HadISST (ts), ERA-Interim (hur), CERES-EBAF (rsut, rsutcs, rsdt) |
| Cox et al. (2018) | ECS | global temperature variability | HadCRUT4 (tasa) |
| DeAngelis et al. (2015) | hydrologic cycle intensification | radiative fluxes and precipitable water | CERES-EBAF (rsdscs, rsdt, rsuscs, rsutcs), RSS (prw), ERA-Interim (prw) |
| Hall and Qu (2006) | snow-albedo effect | springtime snow-albedo feedback values in climate change vs. springtime values in the seasonal cycle in transient climate change | ISCCP-FH (alb, rsdt), ERA-Interim (tas) |
| Massonnet et al. (2012) | YOD | year of disappearance (YOD) of September Arctic sea ice vs. mean sea ice extent or trend in sea ice extent | HadISST (sic) |
| Li et al. (2017) | future Indian summer monsoon precipitation | present-day precipitation over the tropical western Pacific | GPCP (pr) |
| Lipat et al. (2017) | ECS | climatological Hadley cell extent | ERA-Interim (va) |
| Sherwood et al. (2014) | ECS | lower tropospheric mixing index (LTMI) | ERA-Interim (hur, ta, wap) |
| Tian (2015) | ECS | southern ITCZ index, tropical mid-tropospheric humidity asymmetry index | TRMM (pr), AIRS (hus) |
| Volodin (2008) | ECS | difference between tropical and mid-latitude cloud fraction | ISCCP-D2 (clt) |
| Wenzel et al. (2014) | climate-carbon cycle feedback ($\gamma_{LT}$) | long-term sensitivity of tropical land carbon storage to climate warming | NCEP (tas), GCP (nbp, fgco2) |
| Wenzel et al. (2016a) | land photosynthesis ($\beta$) | carbon cycle - $CO_2$ concentration feedback | NOAA station measurements Alaska and Hawaii (co2) |




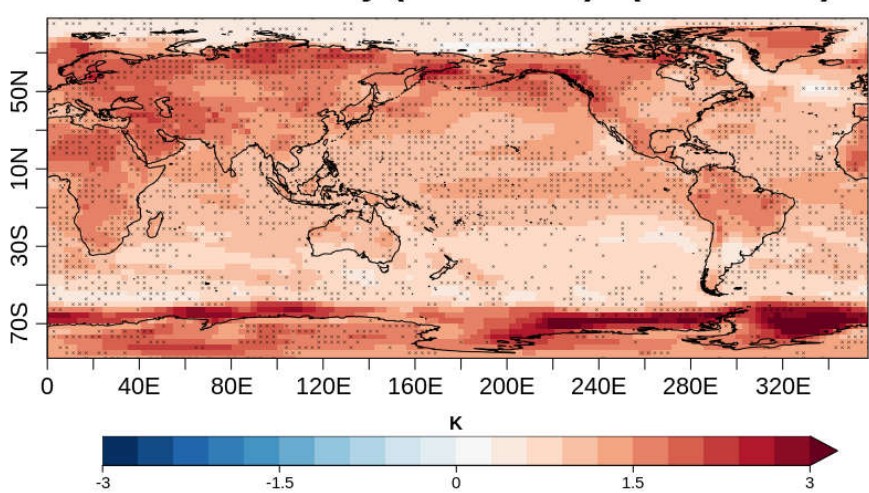

Figure 1 Multi-model mean of projected future June near-surface air temperature anomalies (2006-2099) compared with the period 1961-1990 (colors). Crosses indicate that the 80% of models agree with the sign of the multi-model mean anomaly. The models used in this example are BCC-CSM1-1, MPI-ESM-MR and MIROC5 (r1i1p1 ensembles) for the RCP2.6 scenario. All models have been regridded to the BCC-CSM1-1 grid using a linear interpolation scheme. See Section 3.4.2 for details *on recipe_multimodel_products.yml.*




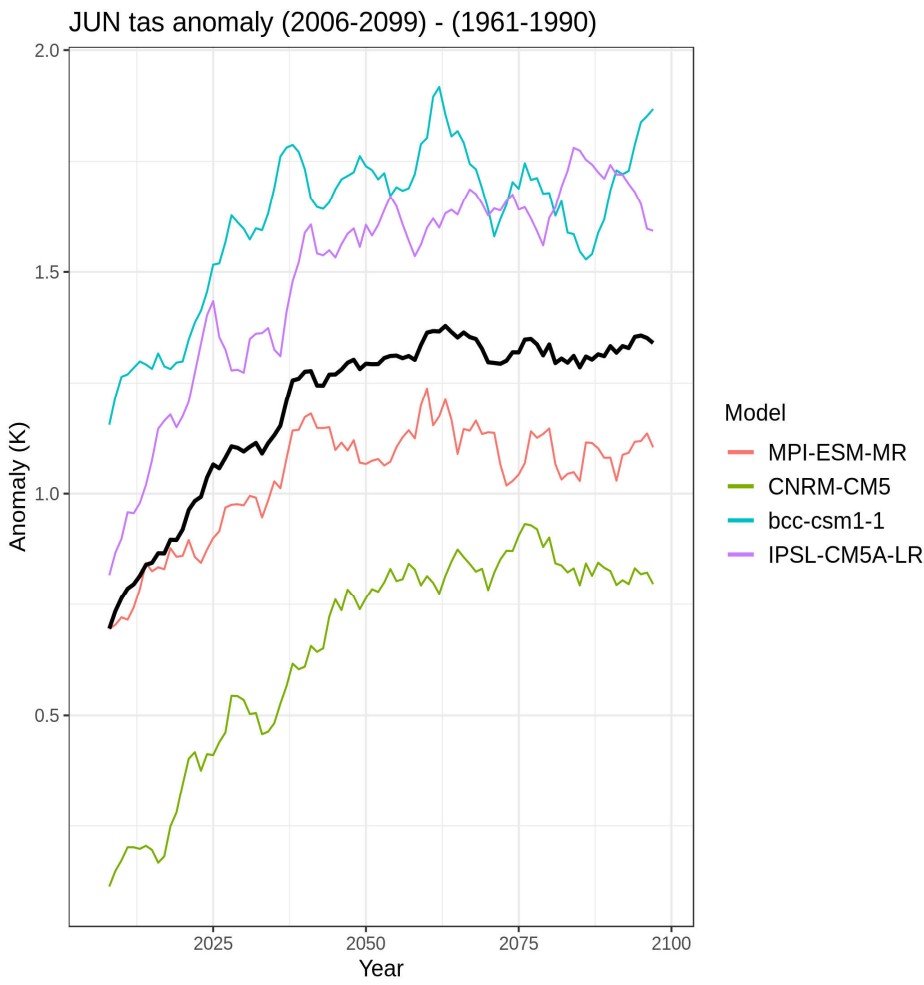

Figure 2 Time series of global average near-surface air temperature anomalies in June for the period 2006-2099 (RCP2.6 scenario) compared to the reference period 1961-1990. The individual models are shown as colored lines, the multi-model mean is shown in black. See Section 3.4.2 for details on *recipe_multimodel_products.yml*.



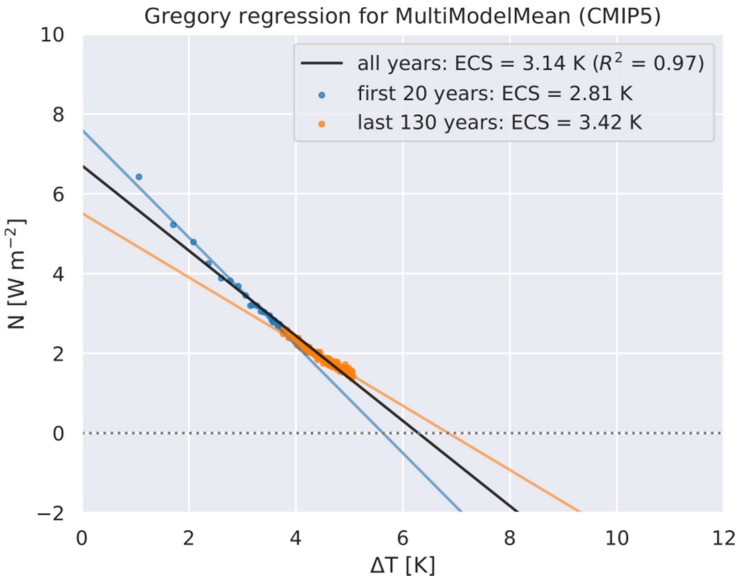

Figure 3 Gregory plot to approximate the effective climate sensitivity (ECS) (Gregory et al., 2004). Shown is the relationship between the differences in global and annual mean top of the atmosphere net downward radiative flux N (W m$^{-2}$) and global and annual mean near-surface air temperature anomalies $\Delta T$ (K) for the CMIP5 multi-model mean. Anomalies are calculated as difference between the abrupt4xCO2 experiment (quadrupling of $CO_2$) and the pre-industrial control run (piControl). The blue dots show the first 20 years of the simulation, the orange dots the last 130 years. A linear regression using only the first 20 years (blue line) instead of all 150 years (black line) results in a stronger feedback (and thus lower ECS). Using the last 130 years only (orange line) results in a weaker feedback (i.e. higher ECS). See Section 3.2 for details on *recipe_ecs.yml*.



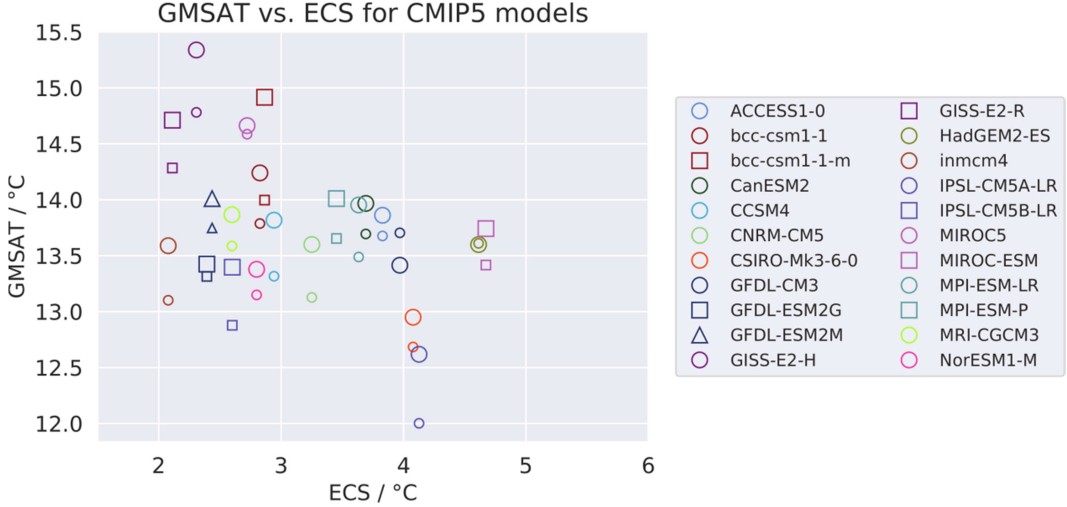

Figure 4 Globally averaged near-surface air temperature (GMSAT) of the historical period 1961-1990 vs. the effective climate sensitivity (ECS) for several CMIP5 models. Similar to figure 9.42a of Flato et al. (2013) and produced with *recipe_flato13ipcc.yml*, see details in Section 3.2.


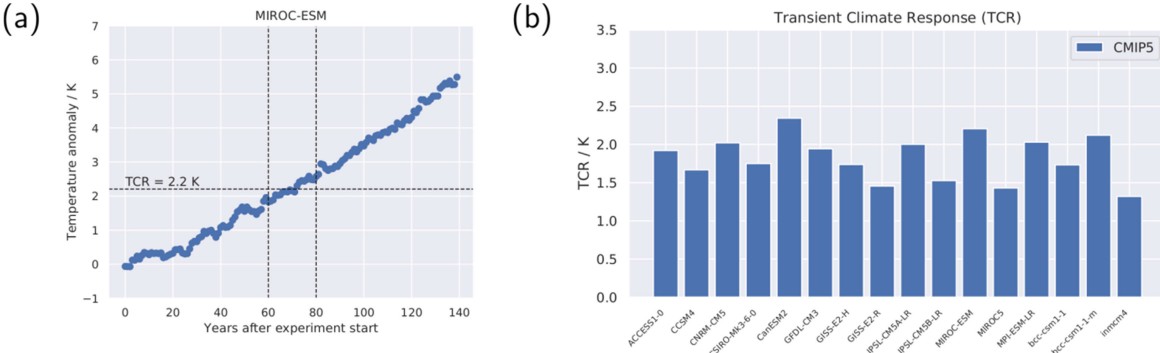

Figure 5 (a) Time series of temperature anomalies from MIROC-ESM experiment 1pctCO2 (1% increase in $CO_2$ per year) compared to the piControl simulation. (b) Transient climate response (in K) for CMIP5 models calculated with the method by Gregory and Forster (2008). For details on *recipe_tcr.yml* see Section 3.2.




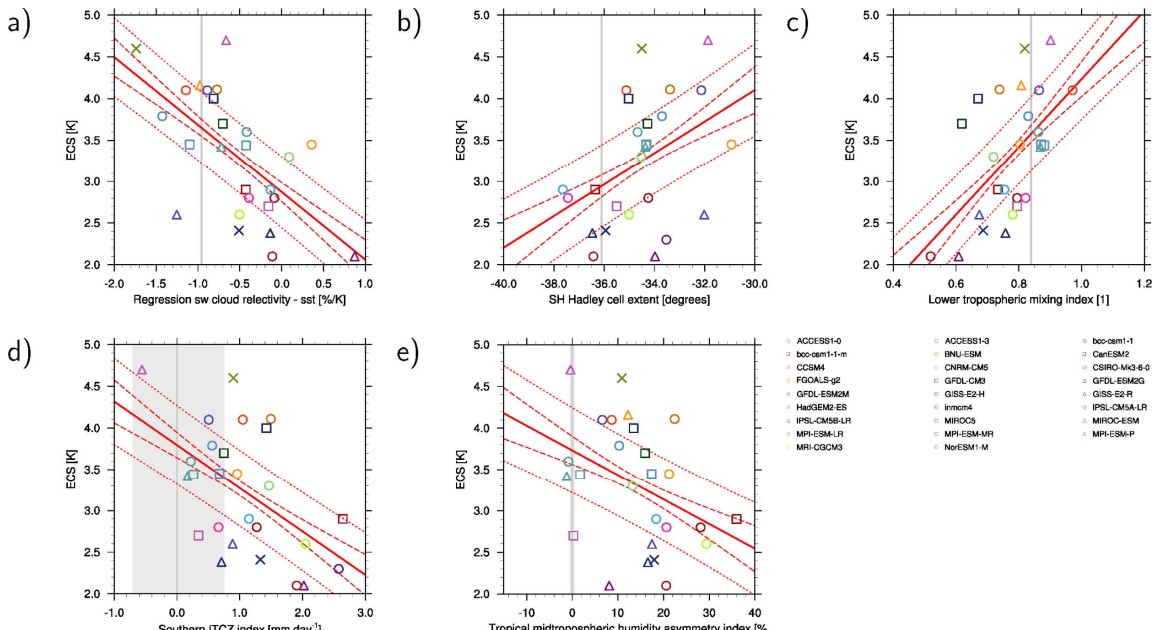

Figure 6 Scatterplots of effective climate sensitivity (ECS) vs. (a) covariance of shortwave cloud reflection (Brient and
Schneider, 2016), (b) Southern Hemisphere (SH) climatological Hadley cell extent (Lipat et al., 2017), (c) lower tropospheric
mixing index (LTMI) (Sherwood et al., 2014), (d) southern ITCZ index (Tian, 2015), and (e) tropical mid-tropospheric
humidity asymmetry index (Tian, 2015) for CMIP5 models (symbols). The vertical gray lines represent the observations, the
shaded areas in light-gray observational uncertainties (if available). The solid red lines represent the regression lines, the
dashed red lines the 25% / 75% confidence intervals of the regression and the red dotted lines the 25% / 75% prediction
intervals of the regression. Similar to (a) figure 6 of Brient and Schneider (2016), (b) figure 4 of Lipat et al. (2017), (c) figure
5c of Sherwood et al. (2014), (d) figure 2 of Tian (2015), and (e) figure 4c of Tian (2015). For details on
*recipe_ecs_scatter.yml* see Section 3.3.1.



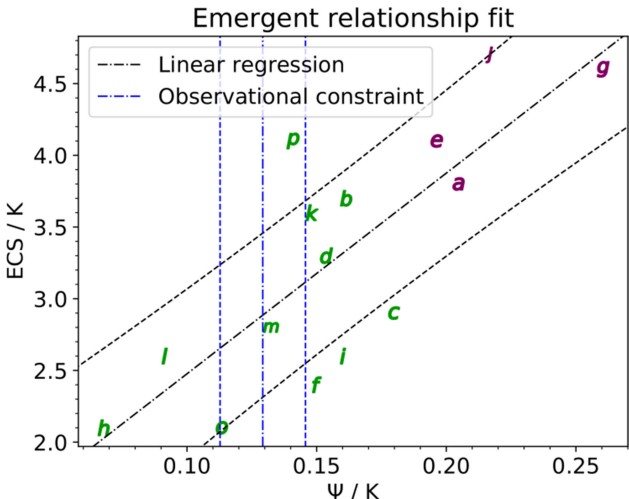

Figure 7 Emergent constraint for effective climate sensitivity (ECS). Shown is the relationship between ECS and the temperature variability metric ψ proposed by Cox et al. (2018). Letters show individual CMIP5 models (for nomenclature details see original publication) with lower sensitivity models in green and higher sensitivity models in purple. The black lines shows the linear fit including the prediction error and the vertical blue lines indicate the observational mean and standard deviation given by the HadCRUT4 dataset. Similar to figure 2 of Cox et al. (2018) and produced *with recipe_cox18nature.yml (*see details in Section 3.3.1.6).



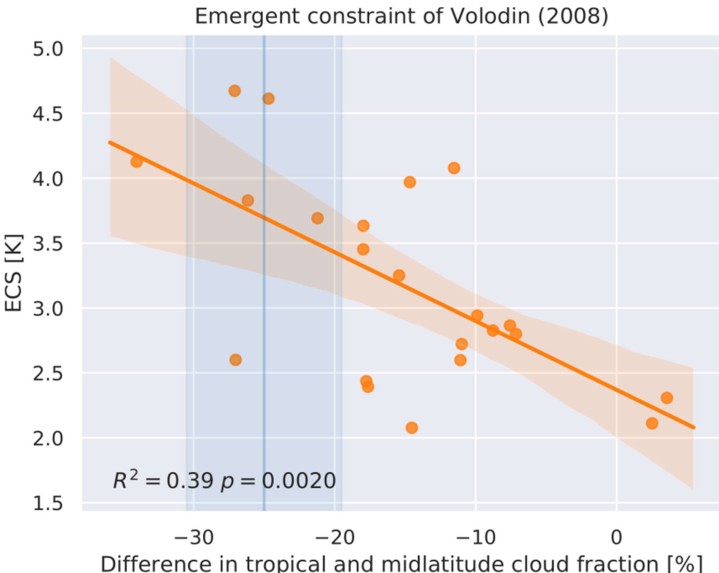

Figure 8 Effective climate sensitivity (ECS) vs. difference in total cloud cover between the tropics (28°S-28°N) and southern mid-latitudes (56°S-36°S) for CMIP5 models (orange dots). The orange line and shaded area show the linear regression line and its 95% uncertainty range (estimated via bootstrapping). Together with the observational estimate (vertical blue line and shaded area), this can be used as an emergent constraint for ECS (Volodin, 2008). The observational range is based on ISCCP-D2 data (Rossow and Schiffer, 1991) and taken from Volodin (2008). Similar to figure 3a of Volodin (2008) and produced with *recipe_ecs_multivariate_constraint_cmip5.yml* (see details in Section 3.3.1.7).

785



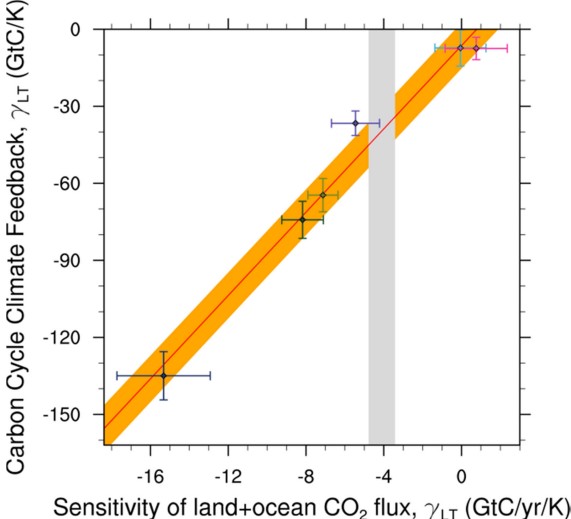

790

Figure 9 Relationship between long-term sensitivity of tropical land carbon storage to climate warming ($\gamma_{LT}$) and short-term sensitivity of atmospheric $CO_2$ to interannual temperature variability ($\gamma_{IAV}$) for CMIP5 models (markers with horizontal and vertical error bars) using the historical simulation. The red line shows the linear regression through the CMIP5 models, the vertical gray area the range of observed $\gamma_{IAV}$. Produced with *recipe_wenzel14jgr.yml*, similar to figure 5a of Wenzel et al. 795 (2014) (for details see Section 3.3.2).

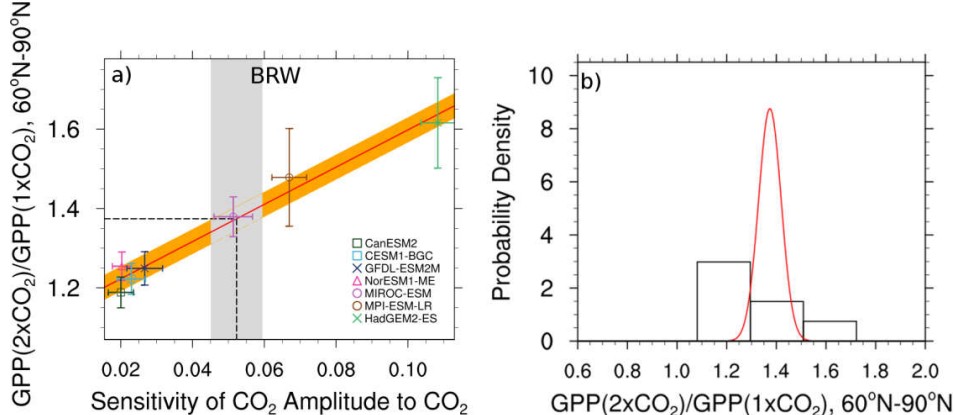

Figure 10 (a) Correlations between the sensitivity of the $CO_2$ amplitude to annual mean $CO_2$ increases at Point Barrow, Alaska (abscissa) and the high-latitude (60°N-90°N) $CO_2$ fertilization on gpp at $2 \times CO_2$. The gray shading shows the range of the observed sensitivity. The red line shows the linear best fit across the CMIP5 ensemble together with the prediction error (orange) and error bars show the standard deviation for each data point. (b) The probability density histogram for the unconstrained $CO_2$ fertilization of gpp (black) and the conditional probability density function arising from the emergent constraint (red). Produced with recipe_wenzel16nat.yml, similar to figure 3 of Wenzel et al. (2016a) (for details see Section 3.3.2).

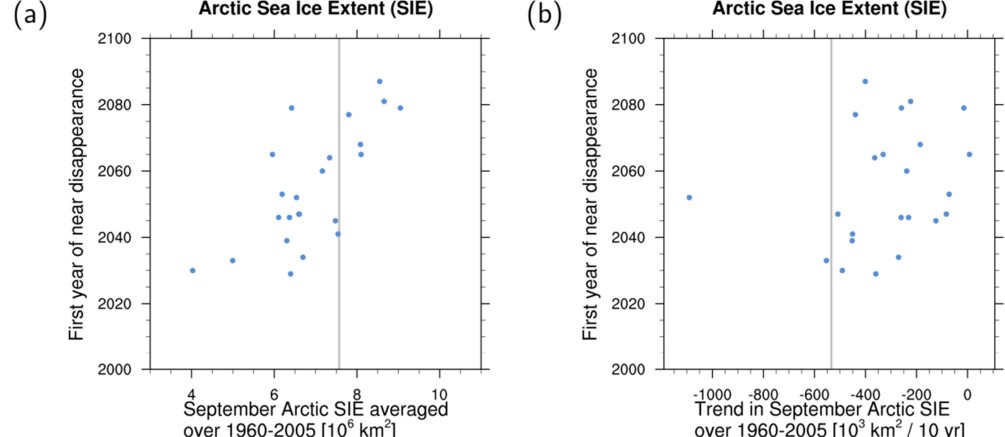

Figure 11 Scatter plot of (a) mean historical (1960-2005) September Arctic sea ice extent (SIE, million km$^2$) and (b) trend in September Arctic sea ice extent (1960-2005) vs. first year of disappearance for scenario RCP8.5. The vertical gray lines are calculated from observations (HadISST, Rayner et al. (2003)), similar to figures 12.31a/d of Collins et al. (2013). For details on *recipe_seaice.yml* see Section 3.3.3.

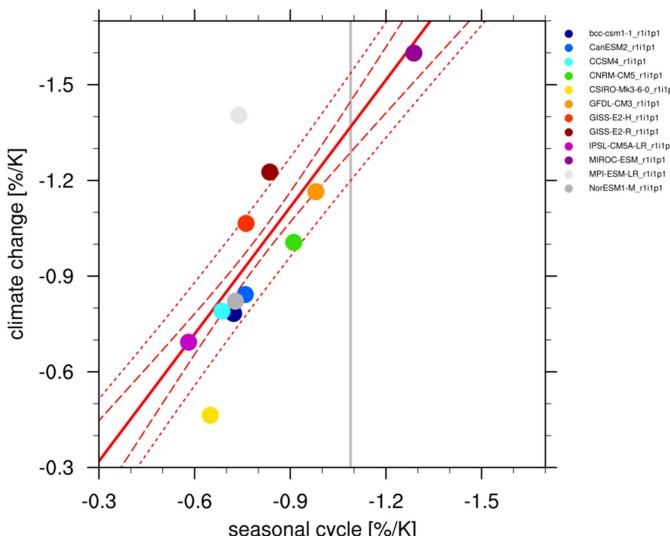

Figure 12 Scatterplot of springtime snow-albedo effect values in climate change (ordinate) vs. springtime $\Delta\alpha_s / \Delta T_s$ values in the seasonal cycle (abscissa) in transient climate change experiments calculated from CMIP5 historical (1901-2000) and RCP4.5 (2101-2200) experiments. The vertical gray line shows the seasonal cycle values calculated from third generation of ISCCP radiative fluxes (ISCCP-FH, Young et al. (2018)) and near-surface air temperature from ERA-Interim (Dee et al., 2011) for the years 1984-2000. Models with higher surface albedos over NH continents poleward of 30°N typically have a larger contrast between snow-covered and snow-free areas, and hence a stronger snow-albedo feedback. Similar to figure 9.45a of Flato et al. (2013), for details on *recipe_snowalbedo.yml* see Section 3.3.4.



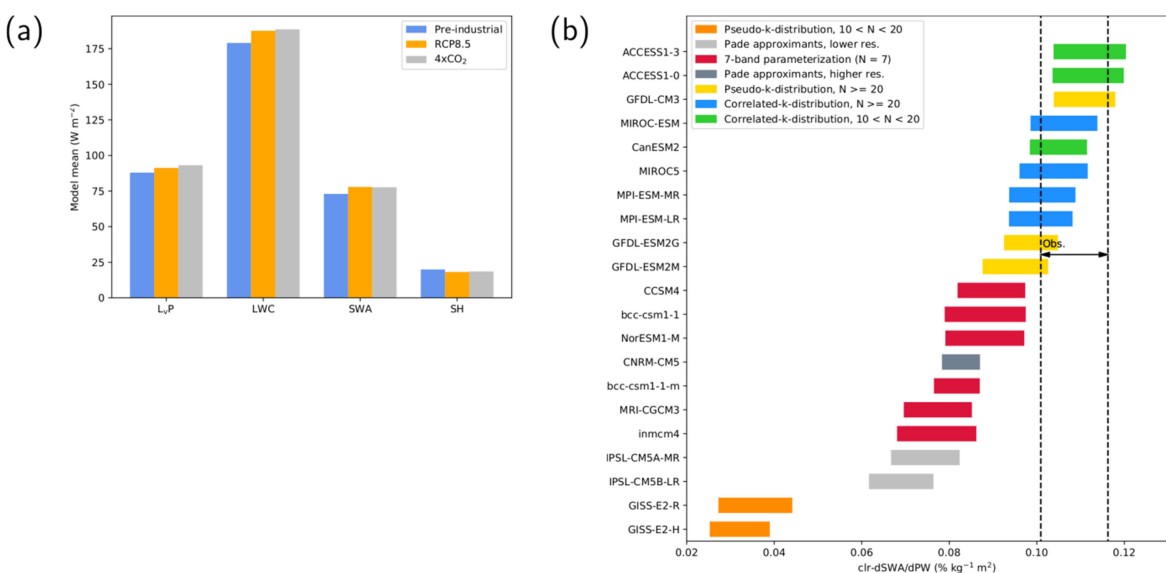

Figure 13 The atmospheric energy budget (DeAngelis et al., 2015): (a) global average multi-model mean sensible heat flux (SH, CMIP5 name hfss) and derived variables LvP (latent heat release from precipitation), LWC (net atmospheric longwave cooling to the surface and outer space calculated as sum of upward longwave radiative flux at TOA and net downward longwave flux at the surface) and SWA (heating from shortwave absorption). The panel shows three model experiments: the pre-industrial control simulation averaged over 150 years (blue), the RCP8.5 scenario averaged over 2091-2100 (orange) and the abrupt quadrupled $CO_2$ scenario averaged over the years 141-150 after $CO_2$ quadrupling in all models except IPSL-CM5A-MR, for which the average is calculated over the years 131-140 (gray). (b) 95% confidence interval for the slope of the regression of clear-sky SWA normalized by the incoming shortwave flux at TOA with the water vapor path (PW, CMIP5 name prw) over the tropical ocean (30°S-30°N), regridded to a 2.5° latitude times 2 kg m$^{-2}$ PW grid for different CMIP5 models (horizontal bars) and for data from CERES-EBAF (Kato et al. (2013); Loeb et al. (2009), SWA) and RSS Version-7 microwave radiometer data (Wentz et al. (2007), PW) together with ERA-Interim (Dee et al. (2011), PW) (dotted lines). The colors indicate different parameterization schemes for solar absorption by water vapor in a cloud-free atmosphere implemented in the models. Similar to figures 1b and 4 from DeAngelis et al. (2015) and produced with *recipe_deangelis15nat.yml* (see details in Section 3.3.5).



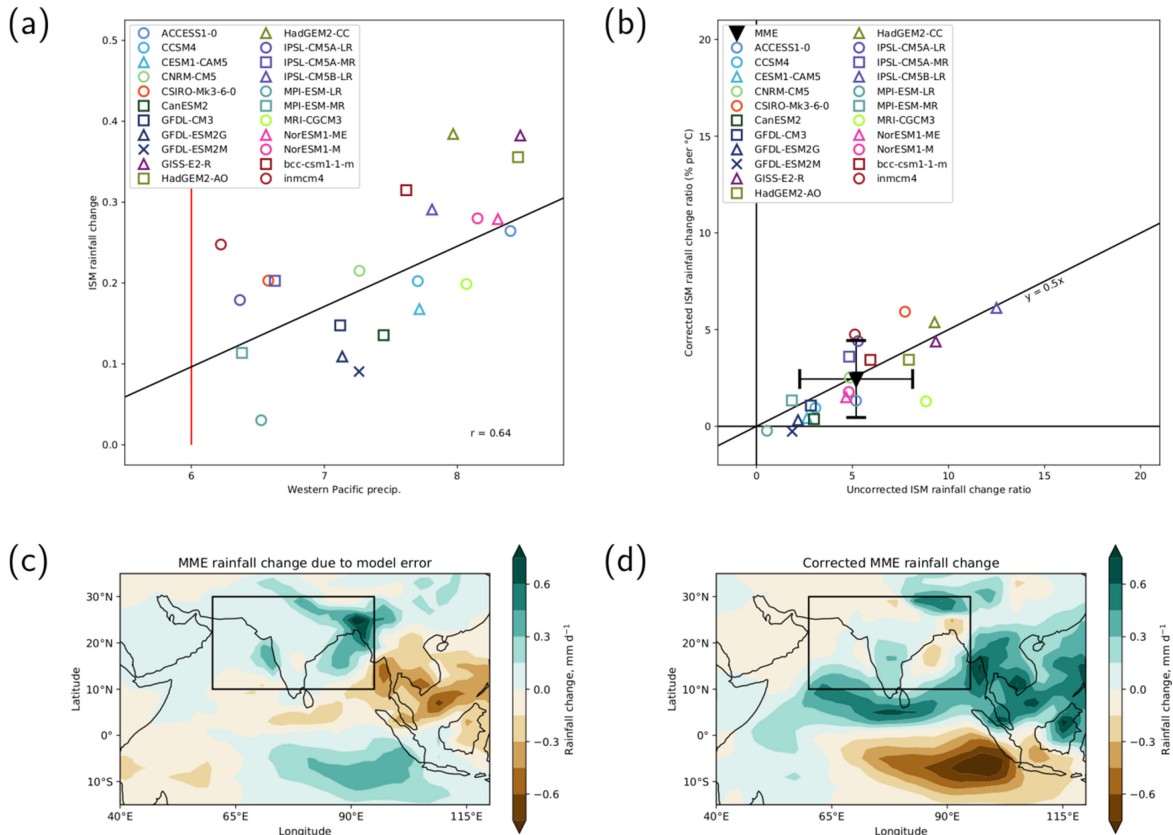

Figure 14 Correction of the Indian Summer Monsoon (ISM, 60°-95°E, 10°-30°N) rainfall projected by models for the RCP8.5 scenario based on the bias in present-day precipitation over the tropical western Pacific (140°E-170°W, 12°S-12°N).
(a) Scatter plot and the linear regression (black line, with the correlation coefficient r) of the western Pacific precipitation (mm day⁻¹) from the CMIP5 historical simulations (1980-2005) and the ISM rainfall change between historical and the RCP8.5 for the years 2070-2099 for different CMIP5 models. The red line indicates the present-day value for the western Pacific precipitation from observations as used in Li et al. (2017) estimated from the Global Precipitation Climatology Project (GPCP) dataset for 1980-1999 (Adler et al., 2003). (b) Uncorrected ISM rainfall change ratio (% per °C) vs. the
corrected ratio from CMIP5 models and the multi-model mean (MME) with the standard deviations shown as error bars. The rain data are normalized by the global mean near-surface temperature change. (c) Projected multi-model mean rainfall change errors and (d) corrected multi-model mean rainfall change over the Indian Ocean. Similar to figure 2 of Li et al. (2017) and produced with *recipe_li2017natcc.yml* (see details in Section 3.3.5).

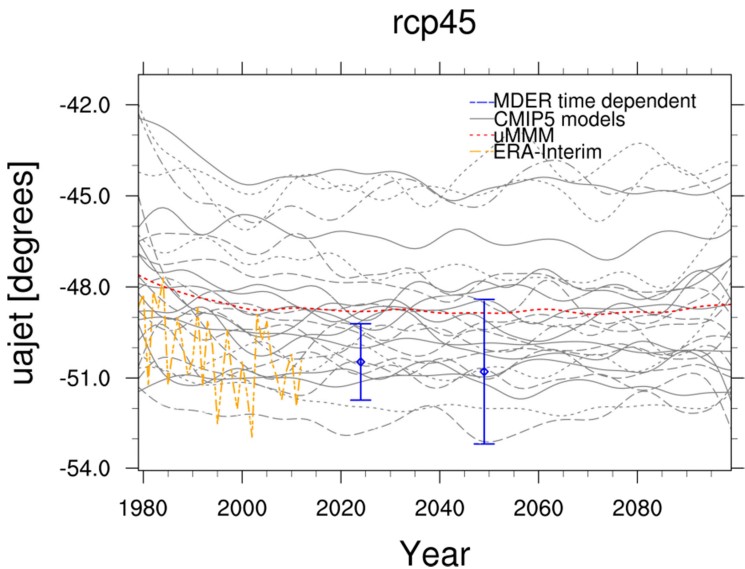


Figure 15 Time series of the austral jet position for the RCP4.5 scenario between 1980 and 2100 based on Wenzel et al. (2016b). The gray lines show individual CMIP5 models and the red dotted line the unweighted CMIP5 multi-model mean. Observationally-based estimates of the jet position from ERA-Interim (Dee et al., 2011) are represented by the yellow dashed line. Blue error bars indicate the predicted jet position by the MDER analysis (Multiple Diagnostic Ensemble Regression) for

the near-term future (2015-2034) and the mid-term future (2040-2059). Similar to figure 5 of Wenzel et al. (2016b) and produced with *recipe_wenzel16jclim.yml*, see details in Section 3.4.1.

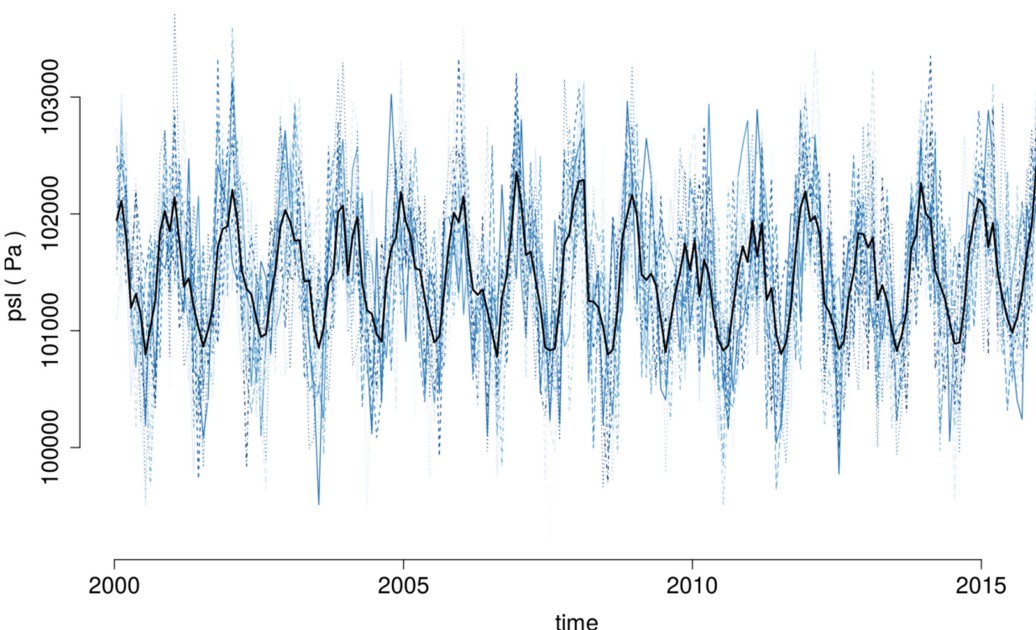

Figure 16 Example of 20 synthetic members of a single dataset ensemble generated by *recipe_toymodel.yml*. Shown are time series of surface-level pressure (psl) averaged over the region 40°E-40°W, 30°N-50°N from 2000-2015 created from monthly mean data from ERA-Interim (Dee et al., 2011). With the user providing an estimate for the standard error e.g. from differences between different observational datasets, this diagnostic can be used to investigate the effect of observational uncertainty. For details see Section 3.4.2.




Figure 17 Time series of global annual mean surface air temperature anomalies (relative to 1986-2005) from CMIP5 models and RCP2.6, 4.5, 6.0, and 8.5 scenarios. The solid lines show the multi-model mean, the shading shows the 5 to 95% range (±1.64 standard deviations). The numbers indicate the number of models these estimates are based on. Similar to Collins et al. (2013) figure 12.5 and produced with *recipe_collins13ipcc.yml* (see Section 3.4.3 for details).


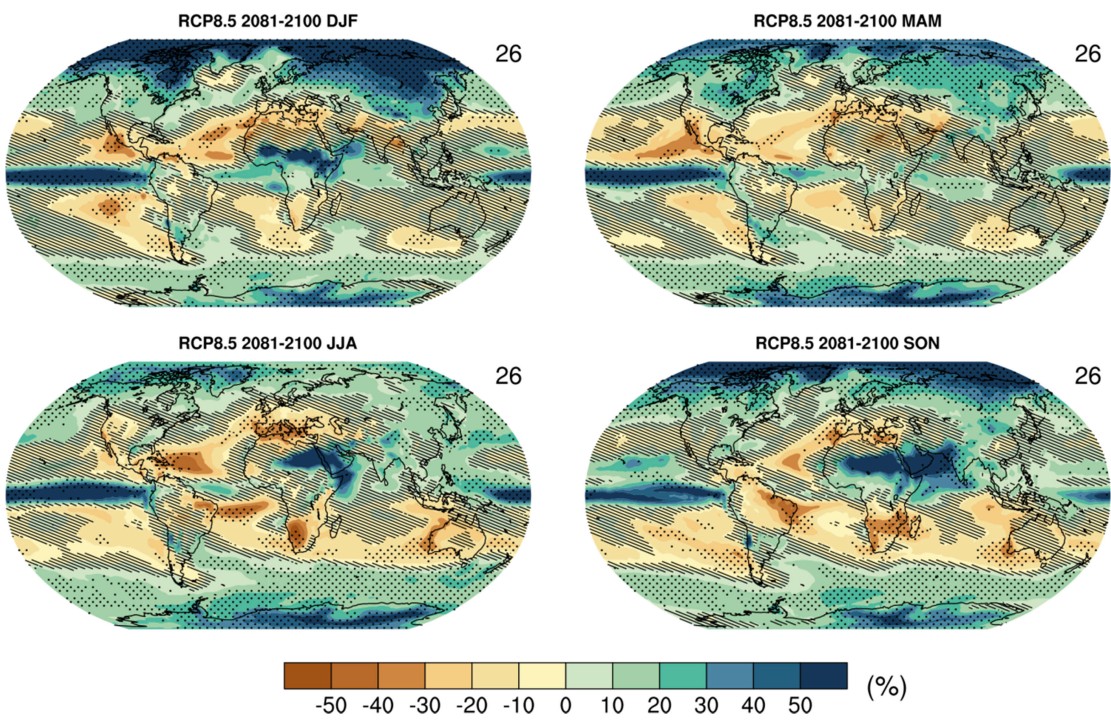

Figure 18 Global maps of seasonal mean change in precipitation from 1986-2005 (reference period) to 2081-2100 for the RCP8.5 scenario. Hatching indicates regions where the multi-model mean change is less than one standard deviation of the internal variability estimated from piControl simulations. Stippling indicates regions where the multi-model mean change is greater than two standard deviations of the internal variability and where at least 90% of models agree on the sign of the change. The numbers in the upper right of each panel indicate the number of models used. Similar to Collins et al. (2013) figure 12.22 but only for one future time period. Produced with *recipe_collins13ipcc.yml* (see Section 3.4.3 for details).



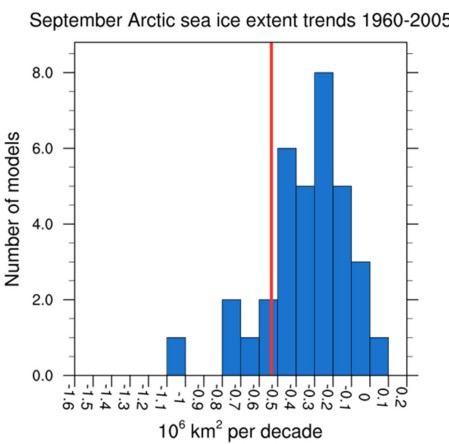


Figure 19 Distribution of trends in Arctic September sea ice extent calculated from the historical simulations (1960-2005) of 26 CMIP5 models (similar to Flato et al. (2013), figure 9.24c). An observational estimate of the trend in summer sea ice extent from HadISST (Rayner et al., 2003) over the same time period is shown by the vertical red line. Produced with *recipe_seaice.yml*, for details see Section 3.4.4.


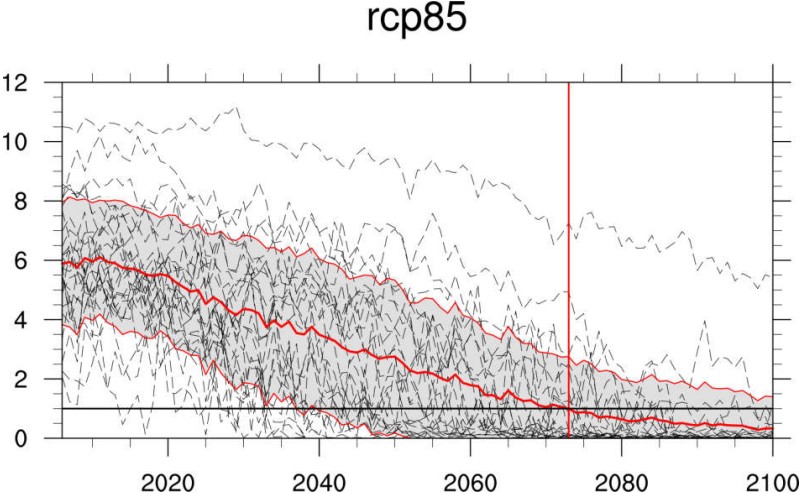

Figure 20 Time series of September Arctic sea ice extent for individual CMIP5 models (gray dashed lines), multi-model mean (thick red line) and multi-model standard deviation (area shaded between thin red lines) for scenario RCP8.5. The year of disappearance (sea ice extent below 1 million km$^2$) obtained from the CMIP5 multi-model mean is indicated by the vertical red line (similar to Collins et al. (2013), figure 12.31e). Produced with *recipe_seaice.yml*, for details see Section 3.4.4.