# Peer review of "Earth System Model Evaluation Tool (ESMValTool) v2.0 – diagnostics for emergent constraints and future projections from Earth system models in CMIP"

_Geoscientific Model Development, 2020_

## Referee Comment (RC1) · Anonymous Referee #1 · 21 Apr 2020

The authors present a description of the latest version of ESMValTool, including details on the new evaluation metrics and 'recipes' that are included. These are clearly linked to the original publications which describe the metrics and examples are provided. This is a clear and well structured paper that I am happy to recommend be published with only minor changes.

I do have two minor comments on the tool and its presentation here. The first is regarding the recipe names which seem somewhat arbitrary. It might be clearer if they followed a standardised format. The other comment is on the various example emergent constraint plots. While it's certainly useful to be able to directly compare with the published work, the very different plot styles jars slightly when presented together like this. Would it be possible to make the original paper formatting of the plots optional, otherwise reverting to a single consistent format? Would it also be possible to add R^2 values to the plots to indicate how well a linear fit really captures the relationship in the models?

Other, minor, grammatical comments: L5: "...implemented include ECS..." -> "...implemented include constraints on ECS..." L195: "used as emergent constraint." -> "used as an emergent constraint."
* * *

---

## Referee Comment (RC2) · Anonymous Referee #2 · 11 May 2020

This paper describes part of the functionality of the ESMval tool that can be used to evaluate and intercompare CMIP (and other) model data. Tools that automate part of the process of collocating and analysing data are immensely useful as they increase efficiency, avoid redundancy and minimise the risk of errors. Given the ever larger flow of (model) data these tools can rightfully be considered part of our modelling toolkit. GMD is an appropriate choice of journal for this paper. The paper has a clear structure and is well written although sometimes short on detail.

The paper states that its aim is "to document and illustrate [] these newly added

[Figure]

ESMValTool "recipes". However, very little information is given to the user on how to use these 'recipes' (do we need to set certain parameters? how does the code find the data? What requirements are there for the data, both models and observations?). Rather the paper seems more an advertisement than a technical document (see https://www.geosci-model-dev.net/9/3093/2016/gmd-9-3093-2016.pdf for an example of the latter). This is not necessarily bad but the paper does not make it easy for users to find the technical documents to obtain this information. As a side note: the readme for the tutorial on github is mostly unpopulated.

Furthermore, I miss discussions of following topics:

- a discussion of the spatio-temporal resolution of the model data and observations used by ESMvaltool. P 3, l 83 states that "any arbitrary model output" can be used. Is that really true? Can I use e.g. both yearly averaged data and hourly data?

- a discussion of the tool's expectation when it comes to the format of observational data. Presumably these should be gridded.

- a description of how ESMvaltool deals with differences in the spatio-temporal resolutions between datasets. This is alluded to in a single Figure caption but should be clearly stated in the paper as part of the tool's functionality. As a side note, it appears the authors believe that observational errors are all stand in the way of model evaluation but there are two other issues. These are 1) differences in spatio-temporal sampling of different datasets and the representativity issues that result (see e.g. https://www.atmos-chem-phys.net/17/9761/2017/ and the references therein); 2) appropriateness of the observation operator (i.e. the model's code that generates a diagnostic that may be compared to observations, e.g. what definition of temperature is used?).

- a discussion of the underlying assumptions in the many regressions used by the tool. My guess is that one important assumption is that individual models can be viewed as independent data points which is unlikely given that often models share (part of) their

code base or at the very least incoporate similar ideas with regards to e.g. sub-grid parametrisations.

- a mention of the graphics formats produced by the tool and whether the user has any control over them.

Finally, I think there may be substantial mistakes in Sect. 3.4.2 that need to be addressed. In addition I found it lacked sufficient explanation.

Minor comments:

Should Table 1 maybe have more information on e.g. the temporal averaging in model data that is needed or do the scripts work with high-freuqency output and perform this averaging themselves?

p. 3, l. 80-85: Can the authors provide references (even if weblinks) for CF-complaincy and CMOR?

p. 3, l. 90: Apparently users can 'import' their own favoruite datasets and use them with ESMValtool. Can the authors provide a brief description of the steps necessary for this to work?

p 5, l 139,140: I do not know whether tas and rlut etc belong to CF-compliant or CMOR definitions but can the authors clarify, also where readers may find further definitions?

p 7, l 189: 'correlation of the covariance'. Shouldn't this just be 'correlation'?

p 13, Sec. 3.4.2: I suggest there is something wrong with either the equation or the definitions here. When alpha=1, the eps_i,m would be drawn from a distribution with imaginary (!) standard deviation (unless beta=0).

There are numerous other issues with this section: - x_i is (probably) not an observation but an anomaly (y has mean 0).

- How is the 'mean correlation between a series of values (x_i,1..M) and a single value

(y_i) defined?

- What is the meaning of eps_i,m? Note: it is also called eps_i sometimes (l. 383), please correct this.

- I'm famliar with the work by Weigel but it seems odd to call alpha the predictability. Don't the random errors eps control the predictability? Probably I misunderstand something but it appears that, beyond the correction of aforementioned errors, this section needs much more explanation.

- the purpose of the toy model is not really explained. I guess it allows the user to put an error estimate on the uncertainty of observations used in emergent constraints etc? Can the user apply this toy model to every constraint or are there limitations? What underlying assumption feed into this toy model? Independent and Identically randomly distributed errors is probably a major assumption and needs to be written down explicitly!

- Toy model may be a confusing choice of word, as the ESMval tool is all about model evaluation. Maybe uncertainty simulator (or estimator) would be a better choice?

p 14, l 409: "including stippling and hatching to indicate significant changes and areas where models do not agree" I found this sentence onfusing. It suggests that stippling/hatching is used to indicate where models do not agree but the caption to Fig 18 states otherwise. Elsewhere in the paper stippling/hatching is used to indicate agreement as well.

p 14, l 412: "where the projections are still uncertain (hatching)." It appears that the use of hatching is quite inconsistent. I understand that the authors are trying to recreate figures found in a large number of papers that are unlikely to be consistent. Maybe this is something to note in the summary or elsewhere, e.g. a 'buyer be ware' clause. After all the authors provide a single tool to generate figures that will be assumed by most users to be consistent in their definitions.

p 15, l 444-446: this explanation of other papers regarding ESMval should be part of the introduction, in my opinion. I would also suggest to add more detail: as a user I want to know which paper to use to find what information.

p 15, Sect 4: I suggest removing the names of recipes. The serve no purpose in this summary.

p 15, Sect 4: The summary should contain a brief mention of data requirements and limitations of the tool. As it stands it is a brief rehashing of the the list of emergent constraints and nothing more.

---

## Author Comment (AC1) · 25 Jun 2020

**Reply to the reviewers' comments on the GMDD manuscript**

Below we address the comments of reviewer #1 raised during the open discussion of the paper "Earth System Model Evaluation Tool (ESMValTool) v2.0 – diagnostics for emergent constraints and future projections from Earth system models in CMIP". We would like to thank the reviewer for the time and effort reviewing the paper. We

feel it has improved thanks to the constructive comments. We have listed all reviewer comments below and our answers are provided in blue. All line numbers refer to the "track changes" version of the revised manuscript that is provided alongside the revised manuscript files.

**Anonymous Referee #1**

The authors present a description of the latest version of ESMValTool, including details on the new evaluation metrics and 'recipes' that are included. These are clearly linked to the original publications which describe the metrics and examples are provided. This is a clear and well structured paper that I am happy to recommend be published with only minor changes.

We thank Reviewer #1 for providing helpful comments to improve the manuscript.

I do have two minor comments on the tool and its presentation here. The first is regarding the recipe names which seem somewhat arbitrary. It might be clearer if they followed a standardised format. The other comment is on the various example emergent constraint plots. While it's certainly useful to be able to directly compare with the published work, the very different plot styles jars slightly when presented together like this. Would it be possible to make the original paper formatting of the plots optional, otherwise reverting to a single consistent format? Would it also be possible to add R^2 values to the plots to indicate how well a linear fit really captures the relationship in the models?

The used naming convention for all recipes that are based on a single peer-reviewed publication or report is recipe_FirstAuthorName_Year_JournalAbbreviation,

e.g. recipe_deangelis15nat.yml. However, for recipes that are based on multiple papers, we relaxed this convention leaving it up to the authors of the diagnostics to decide on a meaningful name, e.g. recipe_seaice.yml combines different diagnostics for sea ice that are based on various articles. An example not fully fitting into either of these categories is recipe_toymodel.yml. In these cases, the naming convention has also been relaxed to any descriptive term chosen by the authors of the diagnostic.

The emergent constraints shown in Figures 6, 7, 8, 9, and 10 have been programmed by different authors (in different languages) as contributions to different projects. In order to give the scientists contributing to the ESMValTool as much freedom as possible and to keep the bar for contributions as low as possible (which is admittedly already quite high) we consider this fine. Homogenizing these figures would require significant recoding. All these diagnostics do, however, output the results as netCDF files, so any plotting program could be used with the ESMValTool output to produce additional plots in the format and layout as desired. Following the suggestion of the reviewer, we added the $R^2$ values to all panels of Figure 6.

Other, minor, grammatical comments: L5: "...implemented include ECS..." -> "...implemented include constraints on ECS..." L195: "used as emergent constraint." -> "used as an emergent constraint."

Changed as suggested.

---

## Author Comment (AC2) · 25 Jun 2020

**Reply to the reviewers' comments on the GMDD manuscript**

Below we address the comments and questions of reviewer #2 raised during the open discussion of the paper "Earth System Model Evaluation Tool (ESMValTool) v2.0 – diagnostics for emergent constraints and future projections from Earth system models in CMIP". We would also like to thank reviewer #2 for the time and effort reviewing the

paper and for the constructive comments. We have listed all reviewer comments below and our answers are provided in blue. All line numbers refer to the "track changes" version of the revised manuscript that is provided alongside the revised manuscript files.

**Anonymous Referee #2**

This paper describes part of the functionality of the ESMval tool that can be used to evaluate and intercompare CMIP (and other) model data. Tools that automate part of the process of collocating and analysing data are immensely useful as they increase efficiency, avoid redundancy and minimise the risk of errors. Given the ever larger flow of (model) data these tools can rightfully be considered part of our modelling toolkit. GMD is an appropriate choice of journal for this paper. The paper has a clear structure and is well written although sometimes short on detail.

We also thank Reviewer #2 for helping us to improve the manuscript.

The paper states that its aim is "to document and illustrate [] these newly added ESMValTool "recipes". However, very little information is given to the user on how to use these 'recipes' (do we need to set certain parameters? how does the code find the data? What requirements are there for the data, both models and observations?). Rather the paper seems more an advertisement than a technical document (see https://www.geosci-model-dev.net/9/3093/2016/gmd-9-3093-2016.pdf for an example of the latter). This is not necessarily bad but the paper does not make it easy for users to find the technical documents to obtain this information. As a side note: the readme for the tutorial on github is mostly unpopulated.

In contrast to the tool documentation example given by the reviewer, the documentation of the new features of the ESMValTool would be (in our opinion) too extensive for a single paper. We therefore decided to split the technical description of the tool and its preprocessor and the scientific description of the new diagnostics and metrics grouped by main application. All of the reviewer's questions regarding data format, directory structure, file names, settings of certain parameters, etc. are covered by our companion paper Righi et al., GMD, 13, 1179-1199, 2020, to which we refer for technical details in this paper. We agree with the reviewer that it should be as easy as possible for the reader to find the technical documents. We therefore we added the following paragraph to the beginning of section 3 (lines 118-124):

> "All diagnostics output one or more netCDF file(s) containing the results of the analysis that are then visualized in the figure(s) created. The file format of the figures can be defined in the user configuration file and includes common formats such as png, pdf, ps and eps. For more details on the technical infrastructure of the tool including accepted data formats, data reference syntax (DRS) used for directory and file name conventions, available preprocessor functions, etc. we refer again to (Righi et al., 2020). Further information can be found in the ESMValTool user's guide, which documents all technical aspects of the tool as well as all available diagnostics, see https://docs.esmvaltool.org/."

The reviewer is correct the README on GitHub is quite brief. This is intentional as we think a very brief description of the main purpose of the tool and a link to the user's guide is probably fine. The idea is that referring to the user's guide instead of duplicating information from the user's guide helps to avoid redundancies and reduces the risk of outdated documentation as everything is in one place.

Furthermore, I miss discussions of following topics:

- a discussion of the spatio-temporal resolution of the model data and observations used by ESMvaltool. P 3, l 83 states that "any arbitrary model output" can be used. Is that really true? Can I use e.g. both yearly averaged data and hourly data?

  The tool itself (i.e. the preprocessor) can indeed handle all time and spatial resolutions that are defined in the CMOR tables for a specific CMIP phase. For CMIP6, for examples, this includes time resolutions from (sub-)hourly to yearly and regular as well as irregular latitude-longitude grids. The diagnostics, however, often expect a certain time resolution or require data to be on a regular latitude-longitude grid or on given pressure levels. This is defined by the diagnostic authors and typically depends on the main aim of the diagnostic.

- a discussion of the tool's expectation when it comes to the format of observational data. Presumably these should be gridded.

  The tool expects all input data including observations to follow the CMOR standard (as outlined in section 2). Typically, such data are stored on a regular (i.e. Cartesian longitude-latitude) grid. The CMOR standard also allows for non-Cartesian longitude-latitude grids if the grid and its mapping parameters are defined. For clarification, we added web links for CF, CMOR and the CMOR tables and definitions used in CMIP6 to section 2.

- a description of how ESMvaltool deals with differences in the spatio-temporal resolutions between datasets. This is alluded to in a single Figure caption but should be clearly stated in the paper as part of the tool's functionality. As a side note, it appears the authors believe that observational errors are all stand in the way of model evaluation but there are two other issues. These are 1) differences in spatiotemporal sampling of different datasets and the representativity issues that result (see e.g. https://www.atmos-chem-phys.net/17/9761/2017/ and the references therein); 2) appropriateness of the observation operator (i.e. the model's

code that generates a diagnostic that may be compared to observations, e.g. what definition of temperature is used?).

Regarding regridding, we added the following paragraph to section 3.1 (lines 129-135):

"For this, all data are regridded to the same horizontal grid. In the example shown in Figure 1, all models are regridded to the grid of BCC-CSM1-1 using a linear interpolation scheme. This task is done by the ESMValTool's preprocessor and defined in the recipe depending on the application and user requirements. The user-definable configuration options include definition of the target grid (e.g. 2.5°x2.5°) and regridding scheme (e.g. linear, nearest, area weighted). Regridding/interpolation of the input data in time is currently not supported. For further details we refer to the ESMValTool user's guide (https://docs.esmvaltool.org/)."

Regarding observational errors, the reviewer has a good point. We therefore extended section 3.4.2 briefly mentioning additional sources of uncertainty when comparing observations to models (lines 450-458):

"We would like to note that in addition to the observational uncertainty itself, also spatio-temporal representativeness of observations plays an important role when evaluating models. Schutgens et al. (2017) showed that such representation errors remain even after spatial and temporal averaging and may be larger than typical measurement errors. In addition, also the calculation method of a quantity to be compared with observations can play an important role. This is, for example, the case when comparing satellite retrievals with model quantities that are not derived the same way. Application of satellite simulators such as the Cloud Feedback Model Intercomparison Project (CFMIP)

Observation Simulator Package (COSP; Bodas-Salcedo et al. (2011))
can help to reduce such uncertainties in model evaluation. Both of
these aspects are not covered by the toy model, that only provides an
estimate for the observational uncertainty itself."

- a discussion of the underlying assumptions in the many regressions used by the
  tool. My guess is that one important assumption is that individual models can
  be viewed as independent data points which is unlikely given that often models
  share (part of) their code base or at the very least incoporate similar ideas with
  regards to e.g. sub-grid parametrisations.

The reviewer is correct that an underlying assumption of the regressions (used for
the emergent constraints) is that the individual models are independent. The re-
viewer is also correct that as some modeling groups provide output from multiple
ESMs and even some ESMs from different modeling groups share components
or code, the models are clearly not independent. Duplicated code as well as
identical forcing and validation data in multiple models is expected to lead to an
overestimation of the sample size of a model ensemble and may result in spuri-
ous correlations.

The original studies proposing the emergent constraints shown here do not ex-
plicitly take into account model interdependency. As the aim of this implemen-
tation was to be able to reproduce the original studies, we did not change this
assumption. As the reviewer has a good point, we added the following paragraph
to section 3.3:

"We would like to note that a limitation of the emergent constraints as
currently implemented into the ESMValTool is that model interdepen-
dency, as in the original studies, is not explicitly taken into account. As
some modeling groups share model components or code the models
are not all independent. Duplicated code as well as identical forcing

and validation data in multiple models is expected to lead to an over-estimation of the sample size of a model ensemble and may result in spurious correlations (Sanderson et al., 2015). As a possible approach future implementations of these emergent constraints could, for example, apply a model weighting based on a model's interdependence (e.g. Knutti et al., 2017) or simply reduce the ensemble size taking into account models only that are above a given yet to be defined interdependence score."

- a mention of the graphics formats produced by the tool and whether the user has any control over them.

  We added the supported graphics formats to the beginning of section 3 (see our answer to the reviewer's first comment).

Finally, I think there may be substantial mistakes in Sect. 3.4.2 that need to be addressed. In addition I found it lacked sufficient explanation.
Minor comments:
Should Table 1 maybe have more information on e.g. the temporal averaging in model data that is needed or do the scripts work with high-freuqency output and perform this averaging themselves?

We corrected the mistakes in Sect. 3.4.2 and added some clarifications and explanations (see our answers to the detailed comments below). Thanks for spotting the mistakes.
All of the diagnostics listed in table 1 expect time series of monthly mean data as input. We added this information to the caption of table 1.

p. 3, l. 80-85: Can the authors provide references (even if weblinks) for CF-complaincy
and CMOR?

We added web links for CF, CMOR and the CMOR tables and definitions used in CMIP6.

p. 3, l. 90: Apparently users can 'import' their own favoruite datasets and use them with ESMValtool. Can the authors provide a brief description of the steps necessary for this to work?

As suggested, we added more details on including non-CMOR-like observational datasets into the ESMValTool to section 2 (lines 99-109):

"Such other datasets that are not available via the obs4mips (https://esgf-node.llnl.gov/projects/obs4mips/) or ana4mips (https://esgf.nccs.nasa.gov/projects/ana4mips/) projects and for which no cmorizing scripts are provided can be used with the ESMValTool in two ways. The first is to write a new cmorizing script using an available one as a template to generate a local copy of reformatted data that can readily be used with the ESMValTool. This typically involves saving only one variable per file and adding meta data such as coordinates (e.g. longitude, latitude, pressure level, time) and attributes (e.g. variable standard and long names, units, dimensions) according to the CMOR standard to the dataset(s). The second way is to implement specific 'fixes' for this dataset in which case the cmorizing is performed 'on the fly' during the execution of an ESMValTool recipe. For details on both methods we refer to the ESMValTool user's guide available at https://docs.esmvaltool.org/en/latest/input.html#observations."

p 5, l 139,140: I do not know whether tas and rlut etc belong to CF-compliant or CMOR

definitions but can the authors clarify, also where readers may find further definitions?

These so-called "standard names" of variables (e.g. tas, rlut, etc) are defined in the CMOR tables read in by the ESMValTool. In addition to web links to the CF and CMOR standards, we also added the following sentence on the CMOR tables to section 2 (lines 91-93):

"These tables read in by the ESMValTool contain the definition of all variables, together with their metadata such as units and standard and long names."

p 7, l 189: 'correlation of the covariance'. Shouldn't this just be 'correlation'?

Thanks for spotting this. Corrected as suggested.

p 13, Sec. 3.4.2: I suggest there is something wrong with either the equation or the definitions here. When alpha=1, the eps$_{i,m}$ would be drawn from a distribution with imaginary (!) standard deviation (unless beta=0).

We thank the reviewer for noticing this lack of information that has been addressed by adding (line 435):

"[...] $\beta$ being limited to the range $0 \leq \beta \leq \sqrt{1-\alpha^2}$."

There are numerous other issues with this section:

- $x_i$ is (probably) not an observation but an anomaly (y has mean 0).

We thank the reviewer for noticing this point. Since the goal of this diagnostic is to simulate single-model ensembles from an observational dataset to investigate the effect of observational uncertainty, the word "observations" is used to distinguish from the model output. It is also used following Weigel et al. (2008) who describe the method.

Given that the recipe does not compute the anomaly itself as an extension of the method described in Weigel et al. (2008), we have modified the parameters of y to cover all possibilities to "y $\sim$ N($\mu$,1)". In the ESMValTool implementation, the user can choose between two options: using the original variable or its anomaly.

- How is the 'mean correlation between a series of values ($x_{i,1..M}$) and a single value ($y_i$) defined?

  It is a property of the toy model described in Weigel et al. (2008): "The average correlation coefficient between the forecast ensemble members and the observations is prescribed by a model parameter $\alpha$.". As a clarification, we added "(see toy model properties described in Weigel et al., 2008)" to bullet point #2 (lines 431-432).

- What is the meaning of $eps_{i,m}$? Note: it is also called $eps_i$ sometimes (l. 383), please correct this.

  Thanks again for reporting this problem. The text has been corrected adding that $\epsilon_m$ is a vector of perturbations and $\epsilon_\beta$ scalar perturbation (line 428):

  "The simulated value xm is obtained by multiplying y by $\alpha$, the predictability of the observation, which is set to 1 in this instance, and by adding a vector of perturbations $\epsilon_m$ and the scalar perturbation $\epsilon_\beta$."

- I'm not familar with the work by Weigel but it seems odd to call alpha the predictability. Don't the random errors eps control the predictability? Probably I

misunderstand something but it appears that, beyond the correction of aforementioned errors, this section needs much more explanation.

We agree with the reviewer that much more explanation could improve this section and we want also to keep the description simple to make users understand the aim of the metric while deeper understanding could be obtained from the main reference Weigel et al. (2008). Therefore, we added the following sentence as a clarification on $\alpha$ and the predictability (lines 436-439):

"Parameter $\beta$ is introduced to control the dispersion. For well-dispersed ensembles, skill is independent of the number of simulations involved, while for overconfident model ensembles, skill grows with the ensemble size. Given that $\beta$ accounts for the dispersion, this approach leads $\alpha$ to represent a measure of predictability (Weigel et al., 2008)."

- the purpose of the toy model is not really explained. I guess it allows the user to put an error estimate on the uncertainty of observations used in emergent constraints etc? Can the user apply this toy model to every constraint or are there limitations? What underlying assumption feed into this toy model? Independent and Identically randomly distributed errors is probably a major assumption and needs to be written down explicitly!

As suggested, we added the list of assumptions made (lines 440-444):

"This toy model is based on very simplifying assumptions: (1) normality and stationarity, the climatology and the ensemble distributions are assumed to be stationary and normally distributed; (2) well-calibrated model climatology, each ensemble member has the same climatology as the observations; (3) stationary skill, spread and correlation do not vary from sample to sample; (4) predictable signal and observational errors, requires the signal to be given by $\alpha x$, and therefore it is determined by the verifying observation (Weigel et al., 2008)."

[Figure]

- Toy model may be a confusing choice of word, as the ESMval tool is all about model evaluation. Maybe uncertainty simulator (or estimator) would be a better choice?

  We would prefer to keep the name of the recipe for two reasons: (1) "toymodel" is the name used in Weigel et al. (2008); (2) the name is already in use by the software of the MAGIC portal (Copernicus Climate Change Service).

p 14, l 409: "including stippling and hatching to indicate significant changes and areas where models do not agree" I found this sentence onfusing. It suggests that stippling/hatching is used to indicate where models do not agree but the caption to Fig 18 states otherwise. Elsewhere in the paper stippling/hatching is used to indicate agreement as well.

In order to clarify the use of stippling and hatching, we reformulated this phrase as follows (lines 471-473):

  "including stippling to indicate large changes with high model agreement and hatching to indicate areas with a small signal or low agreement of models"

p 14, l 412: "where the projections are still uncertain (hatching)." It appears that the use of hatching is quite inconsistent. I understand that the authors are trying to recreate figures found in a large number of papers that are unlikely to be consistent. Maybe this is something to note in the summary or elsewhere, e.g. a 'buyer be ware' clause. After all the authors provide a single tool to generate figures that will be assumed by most users to be consistent in their definitions.

Hatching is used throughout the paper to indicate a small signal or low agreement of models. In order to clarify this, we extended this sentence (lines 476-478):

"This example also shows quite large regions where the projections are still uncertain, i.e. the multi-model mean signal is smaller than one standard deviation of the natural variability estimated from preindustrial control simulations (hatching)."

p 15, l 444-446: this explanation of other papers regarding ESMval should be part of the introduction, in my opinion. I would also suggest to add more detail: as a user I want to know which paper to use to find what information.

As suggested, we moved this paragraph to the introduction (lines 52-59).

p 15, Sect 4: I suggest removing the names of recipes. The serve no purpose in this summary.

Changed as suggested.

p 15, Sect 4: The summary should contain a brief mention of data requirements and limitations of the tool. As it stands it is a brief rehashing of the the list of emergent constraints and nothing more.

Following the reviewer's suggestions, we extended the summary section adding the following paragraph on the ESMValTool's data requirements and limitations (lines 568-582):

"The ESMValTool v2.0 is an open source software tool that has been specifically developed to facilitate evaluation and analysis of Earth system models participating in CMIP. As such, it can process and analyze CMOR compliant

model output and observational datasets with the particular aim to provide traceable and reproducible results, well-documented diagnostics and metrics and an efficient workflow allowing to evaluate models in more depth and more rapidly than it was typically possible in previous CMIP phases. The CMOR standard is, however, quite detailed and implemented in a relatively strict way in the ESMValTool in order to ensure data consistency and to minimize the probability of errors in the data processing. Increasing the flexibility of the CMOR check and automatic fixes of small inconsistencies is a currently ongoing activity and should make the data processing smoother, especially for datasets which are not part of CMIP or any CMIP-Endorsed-Model-Intercomparison-Project (MIP). This means that a certain familiarity with these data standards is required in order to use the ESMValTool. Another limitation is that for license issues, observations cannot be distributed together with the software package. New users are required to download and process observational datasets before being able to use the tool or to have access to a computing center where observational data for the ESMValTool (i.e. cmorized) are already available. We are currently working on automating this process to facilitate the data retrieval and cmorization process."